# Correctness-Guaranteed Code Generation via Constrained Decoding

**Lingxiao Li**
Netflix
Los Gatos, CA, USA
lingxiaol@netflix.com

**Salar Rahili**
Netflix
Los Gatos, CA, USA
srahili@netflix.com

**Yiwei Zhao**
Netflix
Los Gatos, CA, USA
yiweiz@netflix.com

## Abstract

Language Models (LMs) are increasingly being used for code generation, but ensuring the correctness of generated programs remains a significant challenge. Although imperfect code may be acceptable during software development with human oversight, domains such as video games and robotics require one-shot correctness for runtime-critical components. We present a constrained decoding algorithm for generating semantically correct programs that incorporates a context-sensitive parser, which, at each step, outputs a regular expression that satisfies a critical non-extensible property to guide the generation of the next token sequence that can continue to a correct program. To build such a context-sensitive parser, we propose a framework of a dynamic tree of parsers (ToP) during parsing, where each parser corresponds to a modular context-free grammar enriched with contextual information such as variable scopes and type constraints, with tree branches representing ambiguity in the future code segment. We demonstrate our approach through sLua, a strongly typed variant of Lua, showing that our method can generate semantically correct programs conforming to any prescribed scripting API. We further show that, with careful design, our semantic guarantees extend to runtime correctness, as validated in the application of generating game mechanics for a roguelike video game.

## 1 Introduction

The increasing adoption of large Language Models (LM) for code generation has revolutionized software development, particularly in the realm of AI-assisted programming tools such as GitHub Copilot (Dakhel et al., 2023). These tools offer real-time code suggestions and completions, enhancing developer productivity across various domains. However, in runtime critical applications, such as control logic in embodied agents and on-the-fly generative mechanics in video games, since there is no human programmer in the loop, the generated code must be *semantically correct* with *runtime guarantees* in one shot.

Recent advances in constrained decoding have allowed LMs to generate structured data that provably conform to regular expressions (regexes) and more generally Context-Free Grammars (CFGs) (Koo et al., 2024; Ugare et al., 2024; Dong et al., 2024). However, conforming to context-free grammatical rules is far from semantic and runtime correctness. Although it is generally believed that constrained decoding can be extended to ensure semantic correctness, two main challenges remain.

- First, the sequential nature of LMs requires immediate semantic feedback for each token generation in an *incomplete program*, which contrasts with the traditional compiler design that relies on an Abstract Syntax Tree (AST) of a complete program before performing semantic checks.

- Secondly, the atomic building blocks of traditional parsing algorithms are *terminals*, which do not align with the tokens[1] of the LM, resulting in challenges such as token alignment and parser state bookkeeping.

To address the first challenge, we propose a general recipe for building a context-sensitive *Tree of Parsers* (ToP) that flexibly incorporates semantic information and handles ambiguity. This is achieved by maintaining a dynamic tree, each node containing a parser that corresponds to a modular CFG of a language construct (Section 4). From an incomplete program, ToP generates a regex satisfying a crucial non-extensible property (Assumption (A1)) that determines the successive character sequences capable of extending the present program while adequately long to move the parser state forward.

To address the second challenge, we have introduced a constrained decoding method (Algorithm 1) that coordinates a context-sensitive ToP with an LM to produce semantically correct programs that can be parsed by the ToP. At each step, the regex output by the ToP guides the LM in generating the next sequence of tokens, which are then used to advance the parser state, resulting in an incremental generation process. Our algorithm includes a token healing procedure (Fig. 2) to handle misalignment when tokens cross the boundaries of the regexes.

We construct a ToP for a custom language, sLua (Section 5), a strongly typed version of Lua with simplified language features. sLua code can be easily translated to Lua for execution. Although our recipe for building a ToP is language-agnostic, strong typing reduces the need for type inference and the number of tree branches (Theorem 5.1). Combining sLua's ToP with Algorithm 1, we have obtained a constrained decoding algorithm that generates semantically correct sLua code, while preserving the ability to generate all possible semantically correct programs.

As an application, with an off-the-shelf LM, we show how our method can be used to generate new mechanics for a roguelike video game, producing diverse and creative code that conforms to a scripting API (Section 6.1). Moreover, with careful API design and reasonable language feature restrictions, we show that our semantic correctness guarantees can extend to runtime correctness without losing expressiveness (Theorem 6.1). Our approach opens up new possibilities for using LMs in runtime-critical applications, enabling the generation of complex, error-free code within production environments.

## 2 Preliminaries

Constrained decoding (Deutsch et al., 2019) is a popular technique for generating structured output with LMs. At each decoding step, tokens that violate the constraint have their probability set to 0 in the next-token probability vector output by the LM. The masked-out probability vector is then renormalized before sampling the next token. While constrained decoding can distort the distribution (Park et al., 2024), it is widely used in both closed-source (OpenAI, 2024) and open-source tools (Willard & Louf, 2023; Dong et al., 2024) due to its simplicity.

The crux of constrained decoding relies on having a stateful oracle that can efficiently return a mask for the invalid next tokens at each decoding step. When the constraint is a regex (Chomsky, 1956), the oracle can be implemented as a Finite-State Automaton (FSA) that accepts the language, so that the set of valid next tokens can be deduced from the transitions from the current FSA state by indexing (Willard & Louf, 2023) or transducing (Koo et al., 2024). These techniques can be extended to CFGs by using a pushdown automaton (PDA) (Hopcroft et al., 2001) instead of an FSA. However, formal models such as FSAs and PDAs are not powerful enough for context-sensitive parsing, rendering these constrained decoding techniques unsuitable.

Poesia et al. (2022) proposed using a completion engine that takes a partial program and outputs the set of valid tokens that extend the program while incorporating semantic con-

---

[1]Throughout we will use "tokens" to refer to the LM tokens, not to be confused with terminals in a CFG.

straints. However, their completion engine is not incremental, leading to full verification of the program after each token generation. In our work, we identify a core requirement (Assumption (A1)) that leads to incremental constrained generation with token healing (Algorithm 1). Moreover, they target relatively simple programming languages such as SQL and SMCalFlow and do not have a general-purpose solution to handle ambiguities arising from multiple context-dependent rules. The approach of Agrawal et al. (2023) also addresses semantic correctness via constrained decoding, but in a narrower setting that focuses on the language feature of dereferencing an object. One key limitation of their method is that it can incorrectly accept long tokens that start with a valid prefix but end with an invalid suffix. Our approach differs by incorporating a token healing algorithm specifically designed to handle such boundary-crossing token errors, ensuring greater program integrity.

Most relevant to our method, the concurrent work of Mündler et al. (2025) developed a type-constrained decoding approach using prefix automata to reduce compilation errors in TypeScript. Their prefix automata are similar in spirit to our Tree of Parsers (ToP), as both ensure that every reachable state can be extended to a valid final program. Our construction, however, leverages a general-purpose CFG parser generator (Shinan, 2018) to process modular grammar templates enriched with contextual information. This allows us to offload the direct handling of grammar parsing to the generator. In addition, our work focuses on generating code with formal runtime correctness guarantees, ensuring that scripts terminate and execute without errors in a live environment, whereas the goal of Mündler et al. (2025) is to reduce compilation errors and improve functional correctness.

**Context-Free Grammar (CFG)**   A CFG (Aho & Lam, 2022, Sec. 4.2) is described by a set of rules (nonterminals) and terminals. A terminal is a regex or a literal string. A rule is a list of terminals and rules. The language of a CFG is the set of strings that can be generated by starting from the start rule and repeatedly unrolling rules until only terminals remain. A program is syntactically correct if it is contained in the language of the CFG. An example CFG for type specification for sLua is shown in Appendix B.2.6, whose language includes strings like `number`, `(string,boolean)->string`.

## 3   Context-Sensitive Constrained Decoding

**Overview**   We start by presenting our overall algorithm in Algorithm 1, which takes a context-sensitive parser `P` and a language model `L` and generates programs that can be parsed by `P`. We defer the construction of `P` to Section 4 and assume for now that `P` supports the following:

- `has_finished()`: Whether a complete program has been parsed.
- `next_regex()`: Returns a regex satisfying Assumption (A1) to match the next segment.
- `feed_text(s)`: Takes in a segment `s` advances the parser state.

At each step of the generation algorithm (Algorithm 1), we use the regex output of `P` to guide the generation of the next segment of the program that will then be fed to advance the state of `P`. We use a token healing procedure (Fig. 2) to handle the token misalignment issue when a token crosses the boundary between two consecutive regexes.

**Regex Requirement**   When using regexes to successively guide constrained decoding, a main challenge is to construct the regexes in such a way that the LM has the option to stop generation when the regex is matched, so that the parser state will advance. Normally, an LM outputs a special stop token to end the generation. However, here we are using an LM to generate one segment of the program at a time. Since an LM is typically trained on complete programs, the probability of sampling a stop token mid-program is close to zero.

**Example 3.1.** *Suppose that the parser indicates that the next program segment should be an integer and the regex is /[0−9]+/. Since this regex can match digits indefinitely, in order to stop generation, the constrained decoding algorithm must decide when to stop. However, if we modify the regex to /[0−9]+<END>/ where <END> is a set of symbols that signal the end of an integer*

---

**Algorithm 1:** Context-sensitive constrained decoding with token healing

---

**Input:** Context-sensitive parser P, language model L, prompt `prompt`
**Output:** Generated program
```
prog, prefix, prompt_tokens ← "", "", L.tokenize(prompt)
while not P.has_finished() do
    // Modify regex from P.next_regex() to support token healing (Fig. 2).
    r ← re.escape(prefix)+ "(" + P.next_regex()+ ")" + ".*"
    A ← build_DFA(r)                        // build a deterministic finite automaton (DFA) from r
    s, output_tokens ← A.initial_state, []
    while s ∉ A.final_states do              // adaptive rejection sampling to find a valid token
        logits ← L.next_token_logits(prompt_tokens + output_tokens)
        order ← argsort(log(-log(random(len(logits))))- logits)
        for t in order do                               // Gumbel-top-k trick (Kool et al., 2019)
            if A.accepts(s, t) then
                s ← A.transition(s, t)
                output_tokens.append(t)
                break

    output, new_prefix ← "".join(output_tokens), ""
    if A.accepts(output[:-1]) then                     // last token is partially matched
        last_token ← output_tokens.pop()
        output ← "".join(output_tokens)
        new_prefix ← prefix of last_token such that A.accepts(output + prefix)
    new_prog ← (output + new_prefix)[len(prefix):])
    P.feed_text(new_prog)                                           // advance parser
    prompt_tokens ← prompt_tokens + output_tokens
    prefix, prog ← new_prefix, prog + new_prog
return prog + prefix
```

---

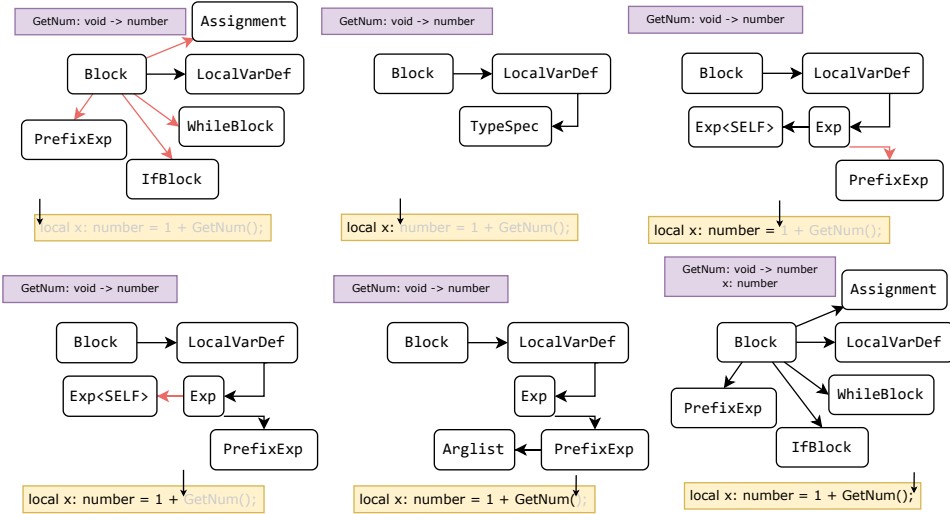

Figure 1: Illustration of the dynamic tree of parsers on an sLua statement. The purple box represents the environment which tracks variable scopes and types. The yellow box shows the parsed program (in black) and the future segments to be parsed (in gray). Rounded rectangles represent parser nodes, and the arrows point from parent nodes to their children. Red arrows indicate parser nodes that will be pruned upon consuming the next segment. Each path from the root to a leaf represents a possible stack of parsing states that could lead to a complete program. See Appendix C.10 for examples of regexes output by the root parser.

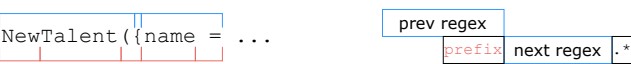

Figure 2: Illustration of token healing for a code segment in the talent script of DCI (Section 6.1). Left: blue (resp. red) brackets indicate boundaries of the regexes from the context-sensitive parser (resp. of LM tokens). For instance, while `({` is a single token, it crosses the boundary between two regexes, one for `NewTalent(` and one for just the curly brace `{`. Right: our token healing method in Algorithm 1 that prepends the matching prefix of the previous token to the regex while allowing the last token to be partially matched.

*depending on the current context, then the **responsibility to stop is delegated to the LM** which is after all the code generator. For example, if the integer is the last argument of a function call, then* <END> *should include* `)`*; if the integer is part of an arithmetic expression, then* <END> *should include any arithmetic operators, and* `)` *if there is a matching* `(` *in the parsed program.*

To formalize intuition in Example 3.1, we impose the following requirement:

**Assumption (A1)** (Non-extensible match)**.** *In Algorithm 1, any regex returned by* `P.next_regex()` *must satisfy: If a string* `s` *matches the regex, then any extension of* `s` *must not match the regex.*

In other words, in Deterministic Finite Automatons (DFAs) constructed in Algorithm 1, one cannot reach a final state from another final state. Therefore, once the DFA is in a final state, we can confidently stop the generation and move on. We defer to Section 4 on how to make the parser `P` satisfy Assumption (A1) using a look-ahead strategy.

**Speeding up with Adaptive Rejection Sampling**   In order to use regex to guide generation, we first build a character-accepting DFA and then use adaptive rejection sampling (Gilks & Wild, 1992) to sample a valid next token at each decoding step. We use adaptive rejection based on the observation that modern LMs generate correct code without constrained decoding in most cases; indeed, adaptive rejection reduces to unconstrained sampling if the first sampled token is valid. The alternative is to compile the character-accepting DFA into a token-accepting DFA (Willard & Louf, 2023; Koo et al., 2024). However, as the regex is constantly changing, the cost of compilation for each semantic segment of the parser is very high. In Table 1, we benchmark the generation speed of the alternatives, showing that the adaptive rejection sampling is more than three times faster than Willard & Louf (2023) and Koo et al. (2024) when used in Algorithm 1.

**Handling Token Misalignment at Regex Boundaries**   Since LMs have custom subword tokenizers, their tokens can cross the boundary of successive regexes output by the context-sensitive parser. To address this misalignment issue, we added a token healing strategy in Algorithm 1 illustrated in Fig. 2. We use a `prefix` variable to keep track of the prefix of the partially matched token from the previous regex, while modifying the regex by prepending the escaped string of `prefix` and appending `.*` to match an arbitrary suffix. This guarantees that the first generated token must start with `prefix` and the last token is allowed to partially match. Thanks to Assumption (A1), after the modification of the regex, if the corresponding DFA is in a final state, then we know the last token is fully or partially matched and we can stop the generation for the regex.

## 4   Context-Sensitive Parsing via a Dynamic Tree of Parsers

Our context-sensitive parsing design is based on two insights:

- It is difficult to inject context into the full CFG of a programming language. However, if we are concerned with only a specific language construct (e.g. a type specification), locally, it is possible to create another CFG that is enriched with semantic context (e.g. Appendix B.2.6) and only accepts semantically correct code.

- Many CFG rules naturally align with the context boundaries. For example, scoping changes when entering blocks, and the context gets updated after processing variable declarations.

Based on these insights, we developed a two-stage solution that builds a dynamic tree of parsers (ToP). Each node in this tree is associated with a CFG and an *interactive* parser (Appendix A.1) generated for that CFG.

The value of our ToP construction goes beyond constrained decoding. It can incrementally verify the semantic correctness of an incomplete program. This can be utilized as a reward model for reinforcement learning to finetune a language model to better produce semantically correct code.

**Stage 1: Refactor the complete CFG into modular CFG templates**

First, we refactor the complete CFG of the target language into modular CFG templates. These templates leave out *slots* that are populated with semantic context during parsing. We add special terminals called *placeholders* to the CFGs to indicate a recursion point where a child parser needs to be spawned if the corresponding placeholder is in the accepting set of terminals of the parent parser. In Appendix B.2, we detail how the complete CFG refactoring is done for sLua.

**Example 4.1.** *Consider a simple modular CFG for string expressions in sLua:*

```
str_exp: str_sum
str_sum: str_atom (".." str_atom)*
str_atom: STRING | "(" str_exp ")" | "<STR_PREFIX_EXP>"
```

*This grammar matches string atoms that can be concatenated using the `..` operator. String atoms can be literal quoted strings (represented by the terminal `STRING` whose regex is omitted), a parenthesized string expression, or a prefix expression of the string type (such as a string variable, a function that returns a string, or a chained function call that eventually returns a string). We use the placeholder `<STR_PREFIX_EXP>` to indicate that upon expecting this terminal to be one of the possible outcomes (in addition to string literals and parenthesized `str_exp`), the parser of this CFG should create a child parser of type string prefix expression (Appendix B.2.7).*

**Stage 2: Define the Tree of Parser Nodes**

Next, to form the ToP, for each modular CFG template from the first stage, we define a tree node type. Each instance of the node contains a parser generated from the corresponding CFG template (Appendix A.1) whose slots are populated with semantic context during parsing. Each node supports three functions, `has_finished`, `next_regex`, and `feed_text`, similar to those described in Section 3 but specialized for the modular CFG associated with the node. Each node manages its own parser state, and the overall context-sensitive parser is simply the root node of the tree.

The parsing process involves the following. Fig. 1 illustrates how ToP evolves when parsing a local variable definition for sLua.

- **Get next regex**: Before forming regexes, a parser node checks whether there are placeholder terminals in its accepting set; if so, a child node will be created for each placeholder. If the accepting set contains terminals that are not placeholders, the parent node spawns a copy of itself as a child node to handle the continuation. To produce a regex for the next valid program segment, if a node has no child, the regex representing the union of all accepting terminals of its parser is returned; otherwise, it forms the union of the regexes from all its children's `next_regex`. This combined regex describes all valid continuations of the current program.

- **Advance parser state**: When a new segment (e.g. sequence of tokens generated by an LM) is fed to a parent node, it delegates the segment to all its children. If any child fails to parse the text indicating a conflict, then that child is removed from the parent. If a child's `has_finished` returns true after feeding a segment, all children of the parent are removed, and the parent parser is advanced with the placeholder of the finished child

(or, in the case of the child being a copy of the parent, the parent's parser state is replaced with the child's).

In Algorithm 2, we provide a pseudocode for a base class for the nodes. At any point, any path from the root node to a leaf represents a stack of nested parsers that *at least one valid complete program can continue from these parser states*. As more code is consumed, the tree branches that result in conflicts are pruned, ensuring that only valid paths remain.

**Environment management**  As in static analysis, we assume that the static environment is being tracked based on the parsed program thus far; the implementation of the tracking is language dependent, for example, as a stack that tracks scopes and symbols in each scope. Before creating a parser node, if its CFG template has slots, we query the current environment and populate the slots accordingly. The environment is then updated when certain parsers finish parsing (e.g. when defining a local variable) or when certain terminals are consumed (e.g. when entering/exiting a scope by encountering do/then/end). For instance, when spawning the string prefix expression child parser in Example 4.1, we need to query the environment for all symbols (variables and functions) that can eventually result in a string type. While ToP has multiple branches to handle future ambiguities, we only need a single environment since it is based on the parsed program.

**Controlling the size of ToP**  The time complexity of parsing with ToP is directly proportional to the size of the tree. While in general this size can be large, when the programming language front loads specifications that reduce ambiguity of parsing (e.g. type specification before any expression), the tree size can be significantly reduced. We characterize the case where the ToP has a "broom" shape, ensuring the tree remains compact. In Lemma B.1, we show that our sLua ToP satisfies this assumption.

**Assumption (A2)** (Broom-shaped ToP)**.** *When a parser node x spawns more than one child, all ancestors of x can have only one child.*

**Satisfying the non-extensible match property**  To ensure that the regex of the ToP root node satisfies Assumption (A1), we recommend a look-ahead strategy that requires the following assumption in modular CFGs:

**Assumption (A3).** *A terminal that is either a placeholder or violates Assumption (A1) in a modular CFG must be followed by non-placeholder terminals satisfying Assumption (A1).*

This assumption is typically met by popular programming languages, which often include many delimiters, such as ; following a statement in C-style languages, end concluding a block in Lua, or ) succeeding a function call in various languages. We prohibit follow-up with a placeholder to prevent recursive look-ahead.

**Look-ahead strategy to extend regexes for non-extensibility**  When a terminal or the regex returned by a child node violates Assumption (A1), we look ahead at the set of next terminals after accepting the terminal or the placeholder and append a regex matching the set of next terminals to the regex. To ensure that modular CFGs do not end with a terminal that breaks Assumption (A1), we add a slot for ending symbols at the end of the starting rule when needed. At instantiation, we query its parent parser for terminals after accepting the child's placeholder and insert a regex representing the union of these terminals to the ending symbols slot. Example regexes obtained after applying this strategy for sLua can be found in Appendix C.10.

## 5   sLua Language and Parsing

We have designed a custom programming language called sLua and developed a ToP for it based on the framework in Section 4. sLua is a strongly typed variant of Lua with restricted language features. It supports common types such as numbers, strings, booleans, functions, and user-defined fixed-size tables. Additionally, it supports variable declarations, standard control flow constructs, expressions, and function calls. In particular, statements must end with a semicolon ; to satisfy Assumption (A3), and a type annotation in

the style of Luau (Roblox, 2019) is required after the declaration of the local variable to ensure satisfaction of Assumption (A2). Dynamic data structures, `nil` pointers, and recursive function calls are not supported in favor of runtime correctness guarantees (Theorem 6.1). See Appendix B.1 for more details and Example B.1 for an example.

In Appendix B.2, we provide an implementation of the ToP for sLua which offers the following linear-time parsing guarantee, as proven in Appendix B.3:

**Theorem 5.1** (Linear-time parsing of sLua). *Consider the implementation of ToP for sLua described in Appendix B.2. Suppose that during parsing the number of variables in the scope, the number of nested function calls, and the number of scopes are bounded by a constant. Then, both the functions* `next_regex` *and* `has_finished` *of the root node in the ToP have a constant time complexity, and the* `feed_text` *function has linear time complexity in the size of the input.*

We selected Lua as our base language because our primary focus is to generate programs conforming to certain API specifications which is achieved by defining global tables in the environment. The embedded nature of Lua makes it particularly well-suited for this purpose. Moreover, LLMs have been trained on substantial amounts of Lua code, so they generalize to sLua well with proper instructions and a few in-context examples.

To run the sLua code, we augment each node of the ToP to output an AST. This involves replacing each placeholder with the corresponding AST of the child node. Based on the AST, we then translate the sLua program to a Lua program (e.g., remove type annotations) and use the Lua interpreter to execute the translated code. The same procedure can be adapted to translate sLua to other host languages.

# 6 On-the-Fly Generative Game Mechanics

## 6.1 Dungeon Crawl Infinite

Recent efforts to integrate LLMs into games at runtime have primarily focused on narrative generation (e.g. Inworld (2021)) or as black boxes (AI Dungeon, 2019) to predict outcomes. We instead consider creating game mechanics on the fly by generating scripts (e.g. Lua) that the game engine can directly run. This approach offers several benefits: direct engine compatibility, consistent and reusable scripts that reduce computational costs, and full transparency provided a way to accurately describe the scripts to the player. The primary challenge, ensuring that the generated code is correct and does not crash the game, is precisely what our work addresses.

We built a turn-based roguelike game called Dungeon Crawl Infinite (DCI) to demonstrate the capabilities of our correctness-guaranteed code generation framework. DCI is directly inspired by Casalini (2012), a highly rated roguelike RPG featuring turn-based combat and advanced character building. In DCI, the player controls a character exploring a dungeon, fighting monsters, and building up their character procedurally with generated *talents* and *effects* given the player's input. See Appendix C.1 for an overview of the game flow. Both talents and effects are implemented as sLua scripts:

- **Talent**: Contains a `Do` function that executes whenever a player or enemy uses the talent.
- **Effect**: Tracks states (e.g., poison stacks) and implements hook functions (e.g., `OnTurnEnd, OnDamageReceived`) that the game engine calls at specific moments.

Both include a `GetDescription` function that returns a dynamic text description reflecting the current game state, informing players about the talent or effect's functionality. Scripts can reference each other through API functions (e.g., `effect.Apply(actor, param)`), enabling complex and synergistic combinations despite the system's simplicity. We design a comprehensive sLua API (Appendix C.3) for DCI containing 55 functions (also extended by each effect definition) that cover rich game mechanics.

To generate talent and effect sLua scripts using Algorithm 1, we have developed specialized root parser nodes (Appendix C.4) on top of the sLua parser nodes to ensure that the generated programs fit the templates for talents and effects. The effect parser node

demonstrates how to easily parse user-defined classes by extending the provided set of sLua parser nodes.

**Provably runtime error-free design**   While determining whether a program in a Turing-complete language executes without errors is NP-hard (as demonstrated by the halting problem), we achieve provable error-free execution by carefully restricting sLua's language features and designing DCI's API. Our approach combines several key strategies detailed in Appendix C.5: a safe callback pattern that eliminates null pointers, guaranteed termination through limited loop iterations and lexical scope function references, and capped recursion depth for effect hooks. This design trades some expressiveness for safety, a reasonable compromise since most game scripts are relatively simple and do not require complex language features. We formalize this below (proved in Appendix C.5).

**Theorem 6.1.** *Given the API in Appendix C.3 and using the techniques described in Appendix C.5, scripts successfully generated for talents and effects in DCI using Algorithm 1 are guaranteed to terminate and execute without runtime errors in the game engine.*

Despite this strong runtime-correctness guarantee, Algorithm 1 itself is not guaranteed to terminate. Nontermination is a well-known issue in constrained decoding; it is part of the more general issue of distribution distortion, where the LM could get the constraints slightly wrong and then corner itself to generate repetition or junk text. We discuss this challenge further in Appendix C.9.

## 6.2   Experiments and Evaluation

To evaluate our pipeline, we propose the following challenging task: generate a *talent category* in DCI. A talent category in DCI consists of 4 talents and up to 4 effects whose theme adheres to a given prompt. Talents and effects can refer to each other in the same category, enabling a complex web of interactions. We use a simple LLM agent to guide the generation of a category (Appendix C.6).

We prepared a total of 8 talent category prompts (detailed in Appendix C.8) to test each method. The effects are generated first, followed by the talents. Generated scripts are registered in the environment and included in future prompts, so that future scripts can refer to previously generated scripts. This extends the correctness guarantees of Theorem 6.1 to talent categories. For in-context examples used in the prompts, we handcrafted 8 effects and 6 talents that cover various API usages.

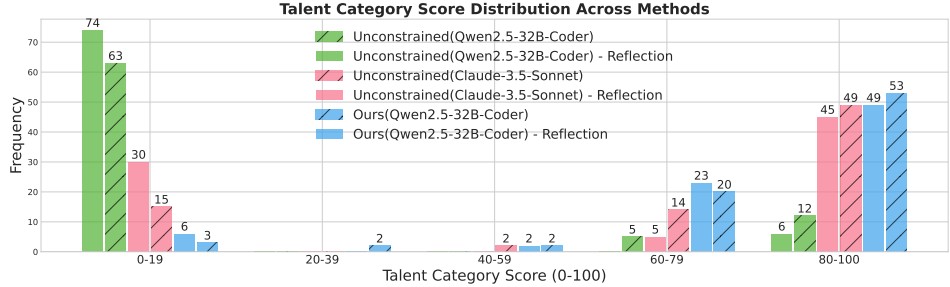

Figure 3: Histogram on talent category scores for various methods. For each of the 8 talent category prompt, we run each method 10 times, for a total of 80 data points for each method in the histogram. An unsuccessful generation is treated as 0 in score. Our method with constrained decoding achieves the highest success rate and quality scores, especially when combined with reflection. For highest quality scripts in score range 90-100, Claude-3.5-Sonnet (with reflection) has 38 data points compared to 33 data points of ours (with reflection). Our method failed only when the generation did not terminate (Appendix C.9).

For baselines, we consider unconstrained generation using Claude-3.5-Sonnet (Anthropic, 2024), one of the best closed-source coding LLMs, and Qwen2.5-32B-Coder (Hui et al., 2024), one of the leading open source coding LLMs. For our method in Algorithm 1, we

|  | Unconstrained | Willard & Louf (2023) | Koo et al. (2024) | Ours |
|---|---|---|---|---|
| tokens/sec | 18.89 | 2.04 | 1.47 | 7.64 |
| compilation/total | 0% | 73.35% | 79.41% | 4.98% |

Table 1: Time comparison of regex-compilation subroutines for constrained decoding in Algorithm 1 for generating a single DCI talent category with prompt "Magic that mixes the power of ice and poison." Measured time excludes the parser update which consistently has a throughput of over 1500 characters per second. All methods first compile a logit processor and then use `VLLM`'s API (Kwon et al., 2023) to carry out constrained decoding. For Willard & Louf (2023), the compilation step involves indexing that converts a character-accepting DFA to a token-accepting DFA. For Koo et al. (2024), the compilation step involves concatenating the DFA of the regex with a character-to-token finite state transducer (precomputed), followed by a determination step using the `OpenFST` library (Allauzen et al., 2007). Note that we did not implement the wildcard handling in Koo et al. (2024), which could explain the speed drop compared to Willard & Louf (2023) as reported in their paper, as their wildcard handling strategy reduces the flexibility of the regexes. The significant speedup of our adaptive rejection sampling is explained by the fact that as the regex is constantly changing, front-loaded compilation to a token-accepting DFA is wasteful compared to the lazy construction of the same DFA of adaptive rejection sampling. This experiment is done with 8 A100 GPUs running Qwen2.5-32B-Coder just like the rest of the experiments.

use the same Qwen2.5-32B-Coder as the LM. In addition, each method includes a *reflection* version. If the initial code generation fails—indicated by parsing error of our parser—then the same LM is prompted to correct the generated code using the error location and the anticipated subsequent regex as guidance in the prompt. This reflection strategy represents the typical rejection sampling technique to use LMs to correct the generated code. For evaluating quality, we use Claude-3.5-Sonnet to output a score between 1-100 to judge the code quality and how well the generated program matches the description returned by the `GetDescription` function for each talent.

The results are reported in Fig. 3. For our method, we terminate the generation and count as failure if the total number of output tokens is over 1500—this typically leads to indefinite repetition of output. In Appendix C.10, we provided examples of failure cases for each method. While unconstrained methods typically failed due to hallucination of variables in scope and incorrect syntax, our method failed only when it did not terminate after the output token limit, when constrained decoding cornered the LM to output an indefinite repetition due to distortion of distribution (Appendix C.9).

In Table 1, we show the inference speed of our method with adaptive rejection sampling compared to regex-compilation alternatives.

## 7    Conclusion

We presented Algorithm 1 to generate correctness-guaranteed programs via constrained decoding with a context-sensitive parser built as a tree of parsers (Section 4). This is demonstrated through sLua, a strongly-typed version of Lua, a roguelike game DCI. As shown by our results and discussed in Appendix C.9, auto-regressive constrained decoding can distort distribution, resulting in nontermination of generation or sub-par quality program quality. A unique opportunity is that the context-sensitive parser can provide additional context information for the LM that can help restoring the distribution. Exploring post-training techniques that use parser hints to combat the distribution distortion issue is a promising future direction for generating high-quality, and correctness-guaranteed programs.

**Acknowledgment**    We thank Alain Dessureaux, Rich Sun, and Danny Diaz for the discussion of on-the-fly code generation in video games and for their feedback on the Dungeon Crawl Infinite game prototype. We also thank Dawen Liang and Nathan Kallus for the discussion on LLM post-training techniques and for their feedback on the initial research plan.

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

## A  Details on Tree of Parsers

We provide in this section our implementation of the ToP framework discussed in Section 4.

### A.1  Interactive CFG Parser

Our tree-of-parsers framework (Section 4) requires a generator of *interactive parsers* (IPs) from modular CFGs during parsing. An interactive parser is a parser that provides the following functions:

- An `accepts()` function that returns the set of terminals (and their regexes) currently accepting.
- A `feed_text(s)` function that advances the parser state by consuming the input string s and raises an error if the input is invalid.

While several parser tools can be used to generate IPs, we use the LALR(1) implementation in Lark (Shinan, 2018) for its linear time complexity and ease of use. While this poses some restrictions on the allowed CFGs, most programming languages can be rewritten in a CFG that belongs to LALR(1).

### A.2  Implementation of ToP

We use an object-oriented approach to define classes for the parser nodes. Recall that each parser node is associated with a modular CFG, and the parser node is only responsible for parsing the program described by its modular CFG. Each parser node class inherits from an abstract base class `BaseParser` that implements `has_finished`, `next_regex`, and `feed_text` — we provide pseudocode (using Python syntax) for these functions in Algorithm 2 — and needs to implement the following fields/functions

- `ip`: An IP for the modular CFG.
- `placeholders`: A set of terminals corresponding to placeholder terminals, indicating when to spawn children.
- `spawn_child(t)`: A function that returns a child parser corresponding to the placeholder t.

- `finish_child(t)`: A function that merges the result of the child parser corresponding to the placeholder `t` back to the parent parser.

Child classes of `BaseParser` may override functions `has_finished`, `next_regex`, and `feed_text` to implement more complex parsing logic. Not included in Algorithm 2 is the lookahead strategy described at the end of Section 4 to make the resulting regex satisfy Assumption (A1). This needs to be applied on a case-by-case basis depending on the specific CFGs and the child parsers.

### A.3 Using ToP for Parsing

We describe in this section the simple algorithm (Algorithm 3) of using a context-sensitive parser with interface described in Section 3 to parse a complete or incomplete program.

## B Details on sLua Language and Parsing

### B.1 sLua Language

Here are the main differences between sLua and Lua:

- Statements must end in `;`. This is to simplify the parsing algorithm to satisfy Assumption (A3) by having natural boundaries between statements.
- The basic types are `number`, `boolean`, and `string`. Custom fixed-size table types (such as `Actor` and `Coord`) are supported similar to named Lua tables. Dynamic data structures and `nil` pointers are not supported in favor of runtime correctness — the scripting API usually has a workaround (e.g. via safe callback patterns as in Example C.1).
- The definition of a local variable or function must fully specify the types using the syntax `local : <type> = <exp>`. For instance,

```
local b: boolean = x == 3;
local foo: (number) -> boolean = function(x) return x < 0; end;
local target: Actor = nil;
local coord: Coord = {x = 3, y = 4};
```

- For functions with non-void return types, all control branches must return with the correct return type.
- Comments (e.g., `--` in Lua) are not supported to simplify parsing. It is possible to support comments by pausing the parsing process and resuming after the comment, but this would complicate the parsing algorithm and can cause tokenization issues.

Strong typing is essential, as knowing the type of an expression before parsing the expression significantly reduces the number of branches in the parsing tree, ultimately leading to our linear-time parsing guarantee (Theorem 5.1).

We feed these specifications about sLua as part of the system prompt whenever we are generating sLua code.

**Example B.1.** *Below are two example sLua scripts for implementing a poisoned effect and a catalyst talent in DCI (Section 6). The poisoned effect inflicts damage to the target at the end of each turn. The catalyst talent doubles the poison count on the target if it is already poisoned, otherwise it inflicts a new poison effect. Variable names prefixed with* `g_` *are global variables. Note all methods are called with* `.` *instead of* `:` *in usual Lua. In our Lua engine, we modified metatables for all involved classes so that* `self` *will be automatically passed as the first argument if what follows* `.` *is a function.*

```
interface ParamData {
    poison: number,
};

do
    NewEffect({
```

---

**Algorithm 2:** Base parser class for tree-of-parsers nodes

---

**Function** `BaseParser.next_regex(self)`
    `accepts ← self.ip.accepts()`
    `exist_nonplaceholder ← false`
    **if** `self.children` is empty **then**
      **for** `t` **in** `accepts` **do**
        **if** `t` **in** `self.placeholders` **then**
          `self.children[t] ← self.spawn_child(t)`
        **else**
          `exist_nonplaceholder ← true`

    **if** `self.children` is empty **then**
      *// We are at a leaf node, so collect regexes from nonplaceholder terminals.*
      `regexes ←` regexes from terminals `accepts \ self.placeholders`
    **else**
      **if** `exist_nonplaceholder` **then**
        *// There are nonplaceholder terminals in addition to placeholders, so we need to make sure there is a continuation of the current parser state.*
        `self.children["<SELF>"] ←` spawn a copy of `self` with a clone of `self.ip`

      `regexes ← []`
      **for** `child` **in** `self.children.values()` **do**
        `regexes.append(child.next_regex())`

    *// Not included in this pseudocode is the lookahead strategy to make the resulting regex satisfy Assumption (A1). This needs to happen before taking the union of the regexes.*
    **return** `"(" + "|".join(regexes)+ ")"`

**Function** `BaseParser.feed_text(self, s)`
    **if** `self.children` is empty **then**
      `self.ip.feed_text(s)`
    **else**
      **for** `t, child` **in** `self.children.items()` **do**
        **try**
          `child.feed_text(s)`
        **except**
          **del** `self.child[t]`          *// prune the errored child parser*

      **if** `self.children` is empty **then**
        **raise** exception; *// dead end*
      **else if** `self.children.keys()== {"<SELF>"}` **then**
        *// All other children failed; merge back to parent.*
        `self.ip ← self.children["<SELF>"].ip`
        `self.children ← {}`
      **else**
        **for** `t, child` **in** `self.children.items()` **do**
          **if** `child.has_finished()` **then**
            *// One child succeeds in parsing; merge back to parent.*
            `self.children ← {}`
            `self.finish_child(t)`
            **break**

**Function** `BaseParser.has_finished()`
    **return** `self.ip.accepts()== {"$END"}`      *// "$END" is a special terminal for the interactive parser to indicate the end of parsing*

---

---

**Algorithm 3:** Parsing a program using a context-sensitive parser

---

**Input:** Context-sensitive parser P, program `prog`
**Output:** (`success, finished`), where `success` indicates whether parsing is
       successful without error, and `finished` indicates whether the the program is
       complete.

$i \leftarrow 0$
**while** not `P.has_finished()` **do**
    `r ← P.next_regex()`                                     // `r` satisfies Assumption (A1)
    `A ← build_DFA(r)`
    `s ← A.initial_state`
    `segment ← []`
    **while** $s \notin$ `A.final_states` **do**
        **try**
            `s ← A.transition(s, prog[i])`
        **except**
            **return** False, False                        // parsing error
        `segment.append(prog[i])`
        `i ← i + 1`
        **if** `i == len(prog)` **then**
            **return** True, False                    // incomplete program
    `P.feed_text(segment)`
`success ← i == len(prog)`  // handle the case where `prog` has extra characters at the end
**return** `success, success`

---

```
        name = "Poisoned",
        beneficial = false,
        detrimental = true,
        OnTurnEnd = function(param)
            param.owner.UpdateHealth(-param.data.poison);
        end,
        OnMerge = function(param, new_param)
            param.duration = g_math.Max(param.duration,
                ↪ new_param.duration);
            param.data.poison = param.data.poison +
                ↪ new_param.data.poison;
        end,
        GetDescription = function(param)
            return "Poisoned, taking " ..
                ↪ g_str.from_num(param.data.poison) ..
                " damage per turn.";
        end,
    });
end

do
    local GetPoisonCount: (Actor) -> number = function(user)
        return 2 + user.GetTalentLevel();
    end;

    local range: number = 5;
    local duration: number = 5;

    NewTalent({
        name = "Catalyst",
        GetRange = function(user) return range; end,
        GetCooldown = function(user) return 12; end,
        Do = function(user)
            return user.WithEnemySelected(range, function(target)
```

```
            local poisoned: boolean = g_effects.poisoned.WhenExists(
                target,
                function(param)
                    param.data.poison = param.data.poison * 2;
                end);
            if not poisoned then
                g_effects.poisoned.Apply(target, {
                    poison = GetPoisonCount(user)
                }, duration);
            end
        end);
    end,
    GetDescription = function(user)
        return "Double the poison count on a target. If the target
            ↪ is not affected by Poison, then inflict " ..
            ↪ g_str.from_num(GetPoisonCount(user)) ..
          " poison.";
    end,
  });
end
```

## B.2   Implementation of ToP for sLua

We describe in this section how to implement the ToP for sLua. Following the framework in Appendix A.2, for each parser node class, we first specify the corresponding modular CFG templates in Lark's grammar from which Lark can generate an IP. The CFG itself contains the set of placeholders, and we further detail how to spawn/finish each type of child parser. All placeholders will be represented using the format <...> in an itemized list, and we use Python's syntax for string formatting to indicate template slots (i.e. {...}). The starting rule in each CFG is always called start. Special treatments unique to each parser node class will be discussed in titled paragraphs.

Here are some comment strategies used across all parser nodes.

**Handling whitespace**   The IPs generated by Lark will ignore all whitespace characters. However, we must explicitly allow whitespace in the regexes returned by next_regex from all parser classes. Hence, we prepend a regex representing any number of whitespace characters to the resulting regex from the root parser. Moreover, whitespace can also be used to extend regexes to guarantee Assumption (A1)—this is handled on a case-by-case basis.

**Caching of IPs**   To prevent rebuilding interactive parsers from identically instantiated CFGs, we use an LRU cache in memory that maps CFG to the IP generated by Lark. We observe that this simple caching strategy has led to a 3x speedup in parsing.

### B.2.1   Block parser

A block in sLua is a sequence of statements. We use the following CFG template for a block:

```
start: block {end_symbols}
block: (stat)*
stat: "<DO_BLOCK>" | "<IF_BLOCK>" | "<FOR_BLOCK>" | "<WHILE_BLOCK>" |
    ↪ "<ASSIGNMENT_STAT>" | "<LOCALVARDEF_STAT>" | "<RETURN_STAT>" |
    ↪ ("<PREFIX_EXP>" ";") | "break" ";"
```

To instantiate this template, we need to populate slot end_symbols, which is a |-separated string of symbols that can end the block. This is used to guarantee Assumption (A3) as recommended by our look-ahead strategy. In most cases, end_symbols is just end. sLua has the following kinds of statements:

- `<DO_BLOCK>`, `<IF_BLOCK>`, `<FOR_BLOCK>`, `<WHILE_BLOCK>`: control flow statements (Appendix B.2.2)

- `<ASSIGNMENT_STAT>`: assignment statement (Appendix B.2.3)

- `<LOCALVARDEF_STAT>`: local variable definition (Appendix B.2.4)

- `<RETURN_STAT>`: return statement (Appendix B.2.5)

- `<PREFIX_EXP>`: expression statement such as function calls (Appendix B.2.7)

- `break` statement to break out of a loop

The generated IP will expect mainly placeholders (except for `;` and `break`). In `spawn_child`, we simply return an instance of the corresponding parser class for each placeholder, and in `finish_child`, we simply feed the placeholder token to advance the IP.

**Ensuring non-extensible match**   Because a keyword (`do`, `if`, `for`, `while`, `return`, `local`) can be a prefix of a variable in scope, the implementation of `BaseParser.next_regex` in Algorithm 1 can break Assumption (A1). For example, a variable might be named `do_it`, and, without careful handling, the resulting regex can match both `do` and `do_it`. To handle this, we add an extra whitespace regex after each of these keywords. In this way, when using the same example, the regex will match `"do "` and `do_it`. Another case is for `end_symbols` to be a prefix of a variable, and we handle this similarly.

**Forcing return statement**   If the block is the top-most block inside a function definition and the function's return type is not void, and not all control flows contain a return statement, then the block must end with a return statement. We detect this situation by tracking the scope of the function definition and whether each control flow branch has returned. If a return statement must be forced, we remove `end_symbols` from the accepting set of terminals of the IP of the block parser, forcing the parser to match a return statement before the block ends.

**Handling break statement**   A break statement is allowed only if there is a while loop or a for loop in one of the parent scopes and there is no function definition in between. We use a similar technique as in how we force the return statement to allow only `break` in the accepting set of terminals if this is the case.

### B.2.2  Control flow parsers

The control flow statements are straightforward to parse. We use the following CFG for each type of control flow parser:

```
do_block: "do" "<BLOCK>" "end"
if_block: "if" "<BOOL_EXP>" "then" "<BLOCK>" ("elseif" "<EXP>" "then"
    ↪ "<BLOCK>")* ("else" "<BLOCK>")? "end"
while_block: "while" "<BOOL_EXP>" "do" "<BLOCK>" "end"
for_block: "for" NEW_VAR_NAME "=" "<NUM_EXP>" "," "<NUM_EXP>" (","
    ↪ "<NUM_EXP>")? "do" "<BLOCK>" "end"
```

Upon encountering each placeholder, the parser will spawn the corresponding child parser.

- `<BOOL_EXP>`: spawn a boolean expression parser (Appendix B.2.11)

- `<NUM_EXP>`: spawn a number expression parser (Appendix B.2.9)

- `<BLOCK>`: spawn a block parser (Appendix B.2.1)

**New variable name terminal**   For `for` loops, the terminal `NEW_VAR_NAME` is a regex that matches any valid variable name. We require variable names to start with an English letter or underscore, followed by the regex `/\w*/`. We require the variable names to be at most 50 characters long.

### B.2.3 Assignment Statement Parser

We use the following CFG for parsing assignment statements:

```
start: assignment_stat ";"
assignment_stat: "<PREFIX_EXP>" "=" "<EXP>"
```

- <PREFIX_EXP>: spawn a *mutable* prefix expression (Appendix B.2.7). The user can specify which fields of a type are mutable, which is incorporated in the type-inference algorithm in the prefix expression parser.
- <EXP>: spawn an expression parser (e.g. Appendix B.2.11) of the target type determined by the preceding prefix expression parser

### B.2.4 Local Variable Definition Parser

We use the following CFG for parsing local variable definitions:

```
start: localvardef_stat ";"
localvardef_stat: "local" NEW_VAR_NAME ":" "<TYPE_SPEC>" "=" "<EXP>"
```

The terminal NEW_VAR_NAME is defined in the same way as in the parser for for loops (Appendix B.2.2).

- <TYPE_SPEC>: spawn a type specification parser (Appendix B.2.6) with available types in scope
- <EXP>: spawn an expression parser (e.g. Appendix B.2.11) of the target type determined by the type specification parser

After the parser finishes, we register in the environment at the current scope the new variable name with the specified type.

### B.2.5 Return Statement Parser

Return statements use the following simple CFG:

```
start: "return" ("<EXP>")? ";"
```

Constructing a return statement parser requires a return type.

- <EXP>: spawn an expression parser of the target type determined by the return type.

If the return type is void, then we remove the placeholder <EXP> from the accepting set of terminals.

### B.2.6 Type Specification Parser

Type specification uses the following CFG template:

```
start: type_spec {end_symbols}
type_spec: base_type | function_type
function_type: "(" (BASE_TYPE ("," BASE_TYPE)*)? ")" "->" type_spec
BASE_TYPE: {base_type}
```

The terminal BASE_TYPE is a regex that matches any valid base type in sLua, including number, boolean, string, void (only for function return type) and table types (Appendix B.2.13). We populate the template slot base_type by querying the current environment when creating the node. In sLua, to simplify the parsing, we allow only one return value from functions. We also disallow nested function types (i.e. function arguments can only be base types) to prevent infinite recursions at runtime (Theorem 6.1).

**Ensuring non-extensible match**    The `BASE_TYPE` terminal can break Assumption (A1) if a base type is a prefix of another base type, e.g. `Actor` and `ActorInfo`. As such, we use the look-ahead strategy described after Assumption (A3) and append the regex after `BASE_TYPE`. We use the same strategy for the other terminals that appear later, such as `NUMBER` and `BOOLEAN`.

### B.2.7  Prefix Expression Parser

A prefix expression is a typed expression that represents field access or function calls, for instance `foo.bar` or `foo(x,y).bar(z,w)`. At creation of the parser node, the target type of the prefix expression must be given. We use the following Lark grammar template for a prefix expression:

```
start: prefixexp {end_symbols}
prefixexp: ("<FIELD>" ("(" "<ARGLIST>" )? ".")* "<FIELD>" ("("
    ↪ "<ARGLIST>")?
```

Note that we do not allow for leading parentheses, e.g., `(foo.bar).boo`. Although this can be supported in the ToP parsing framework, it will slightly break the broom-shaped assumption Assumption (A2) because there will be ambiguity between the prefix expression and the expressions. When constructing a prefix expression parser, as before, we need to provide `end_symbols` to satisfy Assumption (A3). We also need to specify the target type of the prefix expression.

- <FIELD>: upon encountering <FIELD>, the parser will inspect the current environment to identify the set of symbols (variable names, field names, or function names) that can *eventually* lead to a prefix expression of the target type. Then a special type of parser node will be spawned which will essentially be a fixed regex that is the union of all allowed fields. See the paragraph below for details.
- <ARGLIST>: when the preceding <FIELD> is a function, the parser will spawn an argument list parser child node (Appendix B.2.8) to parse the argument list. Note that by design the right parentheses ) are not included in the grammar as they are part of the CFG of the argument list parser—this is to ensure Assumption (A3).

**Tracing symbols that can result in a target type**    To supply the set of symbols when forming the regex for <FIELD> terminal, we use a depth-first search algorithm to trace the type graph and identify a candidate set of types that can result in the target type. This involves tracing through fields types and function return types. Then, while gradually building the prefix expression, we keep track of the type of the current prefix and find fields of the prefix type whose types belong to the candidate set. We allow a few variants, one for searching for only mutable fields (used for assignment statements), and one for searching for function calls (used when the prefix expression is used as a statement as opposed to an expression).

### B.2.8  Argument List Parser

The argument list parser takes in a list of types corresponding to arguments and has the following simple CFG:

```
start: ("<EXP>")* ")"
```

- <EXP>: spawn an expression parser with the target type being the type in that position from the prescribed type list

The accepting set of terminals is modified so that the precise number of arguments will be parsed.

### B.2.9  Number Expression Parser

Next we will show how to build parser nodes for expressions of various types. Since sLua is strongly typed, by the time we need to parse the expression we know what type it should

be, thus reducing the amount of type inference to a minimal level (we still need some inference for the boolean type).

We start with number expressions and show how we can simultaneously exploit the expressiveness of CFG while incorporating context-sensitive information using the ToP framework.

```
start: num_exp {end_symbols}
num_exp: num_sum
num_sum: num_product | num_sum add_op num_product
add_op: "+" | "-"
num_product: num_atom_signed | num_product mul_op num_atom_signed
mul_op: "*" | "/"
num_atom_signed: add_op? num_atom
num_atom: NUMBER | "(" num_exp ")" | "<NUM_PREFIX_EXP>"
```

Here, we omit the definition of the terminal NUMBER, which is a regex that matches any valid number in sLua. Like with a block parser, the CFG template has a slot end_symbols to satisfy Assumption (A3) since the number expression can end in a terminal breaking Assumption (A1).

- <NUM_PREFIX_EXP>: spawn a prefix expression parser (Appendix B.2.7) with the target type being a number

To keep things simple, we do not allow ternary operators in the expression parser. Therefore, the only type that can appear in a number expression is the number type.

### B.2.10 String Expression Parser

The parser for string expressions uses the following CFG template:

```
start: str_exp {end_symbols}
str_exp: str_sum
str_sum: str_atom (".." str_atom)*
str_atom: STRING | "(" str_exp ")" | "<STR_PREFIX_EXP>"
```

The only operator allowed is concatenation, denoted by .. as in Lua.

- <STR_PREFIX_EXP>: spawn a prefix expression parser (Appendix B.2.7) with the target type being a string

### B.2.11 Boolean Expression Parser

Boolean expressions are more complex to parse because it allows numbers and strings to also form boolean expressions. We use the following grammar template:

```
start: bool_exp {end_symbols}
bool_exp: or_exp
or_exp: and_exp ("or" and_exp)*
and_exp: comparison ("and" comparison)*
comparison: bool_comparison | num_comparison | str_comparison
bool_comparison: not_exp (bool_cmp_op not_exp)*
bool_cmp_op: "==" | "~="
not_exp: "not" not_exp | bool_atom
bool_atom: BOOLEAN | "(" bool_exp ")" | "<BOOL_PREFIX_EXP>"
num_comparison: num_exp num_cmp_op num_exp
num_cmp_op: "==" | "~=" | "< " | "> " | "<=" | ">="
str_comparison: str_exp bool_cmp_op str_exp
```

The terminal BOOLEAN is defined as BOOLEAN: "true" | "false". The rules num_exp and str_exp are the same as those in Appendix B.2.9 and Appendix B.2.10, respectively. One sharp bit is that in the num_cmp_op rule, we use literal "< " and "> " instead of "<" and

">" to avoid ambiguity with literals "<=" and ">=" to satisfy Assumption (A1)[2]. As with Lua, all types can be considered as a boolean type and we incorporate this into the type-tracing algorithm for the boolean prefix expression parser.

**Union-type prefix expression**  The set of placeholders for boolean expression parser is: <BOOL_PREFIX_EXP>, <NUM_PREFIX_EXP>, and <STR_PREFIX_EXP>. During parsing, there can be different types of prefix expression parser to spawn at the same time. For instance, suppose that we just parsed an if keyword and are expecting the condition expression which is a boolean. Then all of true, 1 + 1 < 2, and "foo" .. "bar" == "foobar" are valid boolean expressions. One option is to spawn multiple child prefix parsers, one for each placeholder in the accepting set of the IP, but this can be wasteful, as differently typed prefix expressions can share a common prefix string. Instead, we can spawn a union-type prefix expression parser that can handle multiple types of prefix expression and rely on type inference when searching for the set of valid fields. This way, there is only a single child parser being spawned.

**Remark B.1.** *An alternative way to build the boolean expression parser is to include placeholders in the grammar to recursively spawn expression parsers for numbers and strings. While this would simplify the grammar, it would also increase the number of branches in the parsing tree, and the depth of the tree can grow proportionally with the length of the expression.*

**Ensuring non-extensible match**  The field regex can break Assumption (A1) if a field is a prefix of another field, for example, user.power and user.powerups. To handle this, we use the same look-ahead strategy described after Assumption (A3) and append a regex representing the possible symbols after the field name.

### B.2.12  Function Expression Parser

Function expressions in sLua can be a variable of a function type or an inline function definition. We thus use the following CFG template:

```
start: func_exp {end_symbols}
func_exp: func_def | "<FUNC_PREFIX_EXP>"
func_def: "function" "(" (NEW_VAR_NAME ("," NEW_VAR_NAME)*)? ")"
    ↪ "<BLOCK>" "end"
```

Constructing a function expression parser requires the function type (i.e. a type list for all arguments and a return type) to be specified.

- <FUNC_PREFIX_EXP>: spawn a prefix expression parser with the prescribed function type
- <BLOCK>: spawn a block parser (Appendix B.2.1) to parse the function body with an updated scope for the function arguments

### B.2.13  Table Expression Parser

Lastly, an expression can be a fixed-shape table type defined by the user. A table expression can be either a table initialization or a prefix expression that evaluates a table. For table initialization, a field can be required or optional. Here is the grammar template:

```
start: table_exp {end_symbols}
table_exp: "<TABLE_PREFIX_EXP>" | table_initialization
table_initialization: "{" table_init "}"
table_init: (key_eq_value ("," key_eq_value)* ","?)?
key_eq_value: "<KEY>" "=" "<EXP>"
```

Here is how each placeholder is handled:

---

[2]This will require always having a space after "<" and ">" in the code, which is a good formatting style.

- <TABLE_PREFIX_EXP>: spawn a prefix expression parser (Appendix B.2.7) with the target type being a table type

- <KEY>: upon encountering <KEY>, the parser first finds a set of keys that have not yet been used in the table initialization, and forms a regex that matches any of these keys

- <EXP>: spawn an expression parser (e.g. Appendix B.2.11) with the target type being the value type of the key-value pair in the definition of the table type

To make sure that all required keys are present in the table initialization, if not all required keys are present, the parser will not allow the table initialization to end by removing } from the set of accepting terminals.

### B.3 Linear-Time Parsing of sLua with ToP

We now prove Theorem 5.1, that the implementation of ToP in Appendix B.2 gives constant-time functions next_regex and has_finished, and its feed_text has linear time complexity in the size of the input text segment. This implies that when parsing a program of length $n$ using Algorithm 3, the time complexity is $O(n)$.

**Lemma B.1.** *Every node class defined in Appendix B.2 satisfies Assumption (A2).*

*Proof of Lemma B.1.* We will prove by induction on the program length. The base case is trivial, so from now on, we assume Assumption (A2) is satisfied when parsing any shorter program. A node $x$ can be spawned in one of two cases: either when its parent parser has $x$'s placeholder in the accepting terminal set, or $x$ is a copy of the parent parser since there exist nonplaceholder terminals. Hence, it amounts to checking these two cases for all node classes. For clarity, we will use $p(x)$ to indicate the parent node of $x$.

- If $x$ is a block parser (Appendix B.2.1), if $x$ needs to spawn multiple children, it means that we are expecting statements.

  - If the block parser has already parsed some statements, then by induction, right before parsing the previous statement, any ancestor of $x$ had one child. Since the tree of parsers did not change structure between the previous and the next statements, we conclude that, currently that any ancestor of $x$ also has only one child.
  - If the block parser is expecting the first statement, we notice that the placeholder <BLOCK> appears only in the CFG templates of the control flows and the function definition. In these cases, $x$ is the only child of $p(x)$.
    * In the case where $p(x)$ is a control flow parser, since $p(x)$ can only be children of a block parser, we know by induction that when $p(x)$ is spawned, any ancestor of $p(p(x))$ has one child. Upon parsing the first keyword of the control flow (e.g. if), all other children of the block parser $p(p(x))$ will be pruned. Hence, all ancestors of $p(x)$ have a single child.
    * In the case where $p(x)$ is a function expression parser (Appendix B.2.12), by looking at the CFG of $p(x)$, we notice that at the start of the function expression $p(x)$ will have two children, one for function definition and one for function prefix exp. By induction, this means that at that time, any ancestor of $p(x)$ has one child. Since the ancestors of $p(x)$ do not change when $x$ is spawned, we verified that all ancestors of $p(x)$ have a single child.

- If $x$ is any of: control flow parser (Appendix B.2.2), local variable definition parser (Appendix B.2.4), return statement parser (Appendix B.2.5), assignment statement parser (Appendix B.2.3), argument list parser (Appendix B.2.8), and prefix expression parser (Appendix B.2.7), by inspecting the CFG of $x$ and our implementation of accepting terminal sets, we realize only one child parser of $x$ will be spawned at any time. For instance, for prefix expressions, once we parse a field name, we know whether it is a field or a function, and we filter the accepting sets accordingly to disambiguate the rules.

- If $x$ is a type specification parser (Appendix B.2.6), it does not have any placeholder, so $x$ can only be a leaf node.

- If $x$ is an expression parser (Appendix B.2.9, Appendix B.2.10, or Appendix B.2.11), first of all, because of the union-type introduced in Appendix B.2.11, $x$ will have at most one prefix expression child parser. If $x$ needs to spawn both a copy of itself as a child and a prefix expression child parser, we need to verify all ancestors of $x$ have only one child. These are the cases where $x$ can be a child parser:

  - If $p(x)$ is a local variable definition parser, an assignment parser, an if block, a while block, or a for block, then by induction on block parsers, we know when $p(x)$ is spawned, any ancestor of $p(x)$ has one child. We conclude by noticing that $x$ is the old child of $p(x)$.

  - If $p(x)$ is an argument list parser, then $p(p(x))$ has to be a prefix expression parser and both $p(p(x))$, $p(x)$ have a single child. Now $p(p(p(x)))$ has the following possibilities:

    * An assignment statement parser, in which case $p(p(x))$ is the prefix expression on the left-hand side of an assignment statement and $p(p(p(x)))$ has only one child. Again by induction on block parsers, we know all ancestors of $p(p(p(x)))$ have a single child.
    * A block parser, in which case $p(p(x))$ is parsing a function call and we conclude similarly.
    * An expression parser of any type. Thanks to the union-type prefix expression used for boolean prefix expressions (Appendix B.2.11), any expression parser can have at most one prefix expression child parser. Since we are already parsing an argument list inside a function all, the other self-copy child of $p(p(p(x)))$ must have been pruned, so $p(p(p(x)))$ only has one child. Moreover, by induction on expression parsers and the fact that spawning a prefix expression parser child inside any expression parser will also spawn a self-copy child, we know ancestors of $p(p(p(x)))$ only have one child each.

  - If $p(x)$ is a table expression parser (Appendix B.2.13), then the CFG suggests $x$ is the only child of $p(x)$. We conclude that ancestors of $p(x)$ only have a single child each by arguing inductively on the table expression parser $p(x)$.

□

*Proof of Theorem 5.1.* First, since `has_finished` is equivalent to checking whether a special terminal is in the accepting set, it takes a constant time.

Our construction of the modular CFG templates implies that the depth of the ToP during parsing is bounded by the number of nested function calls and the number of scopes (i.e. nested blocks). Since both numbers are assumed to be bounded, the depth is also bounded by a constant. By Lemma B.1, the ToP will be in a broom shape, so its size will be bounded by its depth multiplied by the maximum number of children of any node. The maximum number of children of a node is bounded by the number of its placeholders in its CFG plus one. The block parser (Appendix B.2.1) has the most number of placeholders (8), so we conclude that the total size of ToP is bounded by a constant.

To analyze the time complexity of `next_regex` of the root node, note that it iterates over ToP, spawning children and collecting regexes in the process. The cost of spawning a node is dominated by creating an interactive parser (Appendix A.1) from its CFG. Since variable names are at most 50 characters long (see the end of Appendix B.2.2) and the number of variables is bounded, the size of each instantiated CFG is also bounded by a constant, so the creation of parser nodes takes constant time. Collecting regexes requires scanning over accepting terminals, the number of which is also bounded by a constant. Hence `next_regex` has a time complexity proportional to the size of ToP, which is constant.

For `feed_text`, since we use the LALR(1) implementation of Shinan (2018), advancing each interactive parser takes a linear time in the input text. Since we at most feed the same text to all nodes in the tree which has constant size, overall the `feed_text` function of the root note takes a linear time in the input text segment. □

## C    Details on Dungeon Crawl Infinite

### C.1    Game Flow of DCI

We briefly describe what a typical game flow looks like for DCI.

The player starts the game by generating an adventure given a text prompt (left of Fig. 4). This generates the entire adventure (see Appendix C.7 for details) and the process takes 5-10 minutes. The adventure includes the player's starting talent category, the enemies' talent category, the enemy types, and all kinds of narrative text. Afterwards, the player enters the game world (right of Fig. 4). The game world consists of three levels, with progressively more challenging enemies. The main quest of the game (left of Fig. 5) is to defeat the final boss located at the last level. The game combat is turn-based and each level is a grid (right of Fig. 5). The main progression of the game is through evolution in which the player sends EXP (obtained by defeating enemies) to either improve their stats, level up their talents, or evolve new talents (and the accompanied effects). Talent evolution (Fig. 6) triggers on-the-fly code generation given a player's text prompt, and the player can choose from one of the three generated talents.

All the game UI follows a minimalist design, with texts replacing textures whenever they would normally be needed (e.g., each actor is a moving text).

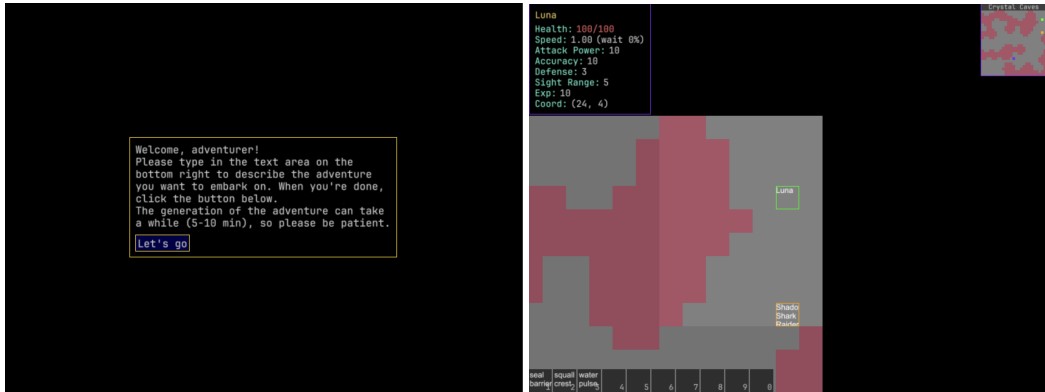

Figure 4: Left: starting screen for DCI before entering an adventure prompt. Right: generated world after adventure generation is finished given the prompt "an aquatic adventure as a seal warrior."

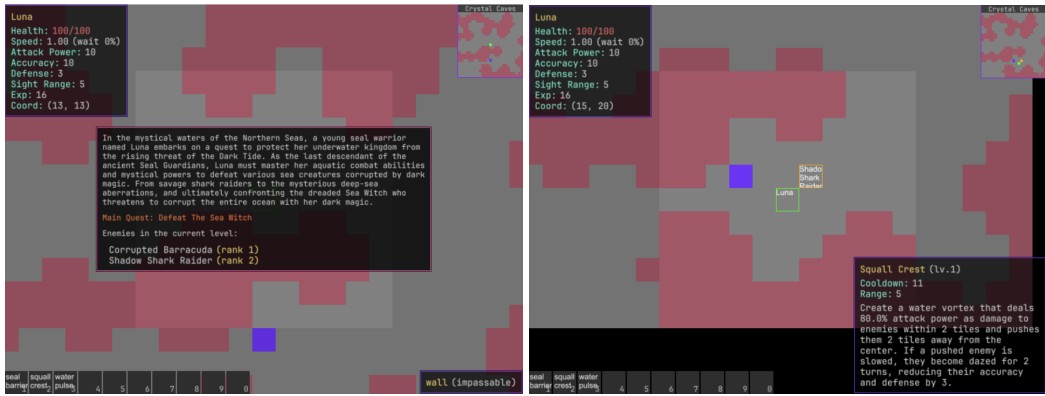

Figure 5: Left: quest panel that shows the story, the main quest, and a list of enemies in the current level. Right: example of a talent "Squall Crest" during a combat with an enemy "Shadow Shark Raider".

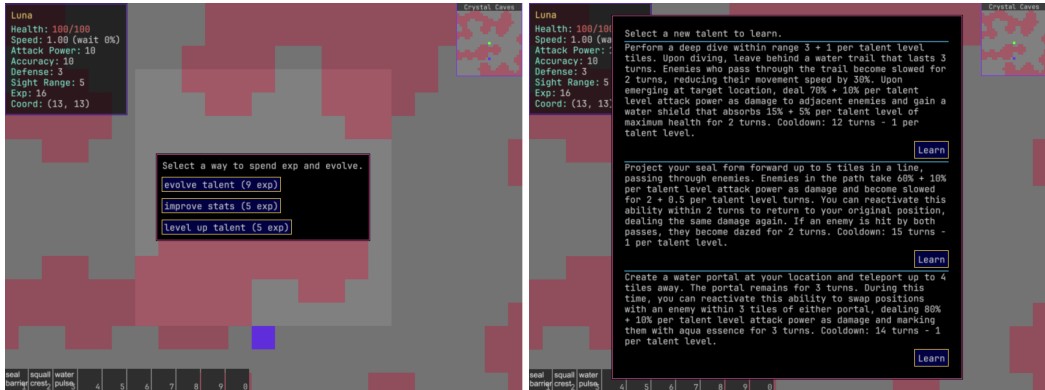

Figure 6: Left: character progression screen. Right: candidate talents to evolve given a text prompt "mobility or teleport".

## C.2 Implementation Details

Our implementation of DCI consists of a Python game server backend and a web-based client interface. The backend hosts individual Lua runtime environments for each session using the `lupa` package[3], while the frontend is built with PixiJS[4]. This architecture enables the web client to receive visual updates from the server and transmit keyboard input back to the game environment, with game logic executed in Lua code that interfaces with Python.

To support our constrained decoding algorithm, we deployed a dedicated LLM server running Qwen2.5-32B-Coder using VLLM (Kwon et al., 2023) on an AWS P4 instance with 8 A100 GPUs. The implementation of Algorithm 1 leverages `outline` (Willard & Louf, 2023) for the indexing step and `irregular`[5] to construct finite deterministic automata (DFA), enabling efficient constrained generation of syntactically valid code.

## C.3 API for DCI

Below is the scripting API for DCI. The implementation of the API in the game engine ensures that regardless of the game state, as long as the API call sites type check, the API call will terminate (in case of a function being an argument, assuming the callback function terminates) and without runtime error. We fed the verbatim API documentation to the LLM when generating the game mechanics scripts.

```
GlobalEffectTable:
  doc: |
    Global table for storing effects. Use `g_effects.<id>` to access the
        ↪ effect by id.
    An effect has three methods:
    - g_effects.<id>.Apply(target: Actor, data: ParamData, duration:
        ↪ number) -> void
      Apply the effect to the target with the given data and duration.
      Type ParamData will be specialized to the effect with <id>.
    - g_effects.<id>.WhenExists(target: Actor, fn: Param -> void) ->
        ↪ boolean
      Apply `fn` to the effect param if it exists. Return true if the
        ↪ effect exists, and false otherwise.
      Type Param always contains
        - duration: number, duration of the effect
        - owner: Actor, the actor that receives the effect
```

---

[3] https://pypi.org/project/lupa/
[4] https://pixijs.com/
[5] https://github.com/MegaIng/interegular

```
        - data: ParamData specialized to the effect with <id>
    - g_effects.<id>.Remove(target: Actor) -> boolean
      Remove the effect from the target if exists. Return true if the
        ↪ effect is removed, and false otherwise.

GlobalGame:
  doc: "The global game table."
  fields:
    level: "Current level."
    ResolveHit: |
      (target: Actor, damage: number, source: Actor) -> void
      Resolve the event where `source` actor hits `target` actor for
        ↪ `damage`.

GlobalMath:
  doc: "Math library. Do not use math functions not present in this
    ↪ table."
  fields:
    Random: |
      () -> number
      Get a random float between 0 and 1.
    RandomInt: |
      (low: number, high: number) -> number
      Get a random integer in [low, high-1].

GlobalString:
  doc: |
    String library. Do not use string functions not present in this
      ↪ table. Use ".." to concatenate strings.
  fields:
    from_num: |
      (num: number) -> string
      Convert a number to a string rounded to the nearest integer.
      It is by design to never have a decimal point in the string.
      To represent a percentage, multiply the number by 100 before
        ↪ calling this function.

Coord:
  doc: "2D coordinate {x: number, y: number}."
  fields:
    x: "X coordinate."
    y: "Y coordinate."
    Add: |
      (other: Coord) -> void
      Add the other coordinate to this coordinate.
    Subtract: |
      (other: Coord) -> void
      Subtract the other coordinate from this coordinate.
    Length: |
      () -> number
      Get the length of the coordinate.

Level:
  doc: "A game level."
  fields:
    MoveActor: |
      (actor: Actor, coord: Coord) -> boolean
      Move the actor to the given coordinate. Do nothing if the
        ↪ coordinate is not passable.
      Return true if the actor is moved, and false otherwise.
    WithActorAt: |
      (coord: Coord, fn: (actor: Actor) -> void) -> boolean
      Apply `fn` to the actor at the given coordinate if there is one.
    GetPushedCoord: |
      (source: Coord, target: Coord, distance: number) -> void
```

```
      Get the coordinate if `source` pushes `target` by `distance`.
    WithRandomEmptyCoordInRadius: |
      (coord: Coord, radius: number, fn: (coord: Coord) -> void) ->
          ↪ boolean
      Apply `fn` to a random empty coordinate in `radius` of `coord` if
          ↪ there is one.
    ProjectRandomActors: |
      (coord: Coord, radius: number, fn: (actor: Actor) -> boolean) ->
          ↪ void
      Apply `fn` to actors in random order in `radius` of `coord`.
      If `fn` returns false, stop the iteration.
    ProjectBall: |
      (target: Coord, radius: number, fn: (coord: Coord) -> void) -> void
      Invoke `fn` for each coordinate in the ball with `target` as the
          ↪ center and `radius` as the radius.
      This is usually used for area of effect calculations.
    ProjectLine: |
      (from: Coord, to: Coord, fn: (coord: Coord) -> void) -> void
      Invoke `fn` for each coordinate in the line between `from` and
          ↪ `to`.

Actor:
  doc: "A game actor."
  fields:
    coord: "Current coordinate."
    health: "Current health."
    max_health: "Maximum health, in range 10-500."
    faction: "Faction of the actor, either 'good' or 'bad'."
    accuracy: |
      Accuracy of the actor, in range 0-20
    defense: |
      Defense of the actor, in range 0-20.
      Chance of hit is 0.5 + 0.05 * (accuracy - defense).
    attack_power: "Attack power of the actor, in range 1-50."
    speed: "Speed of the actor. 1.0 is the default speed, and 2.0 is
        ↪ twice as fast. Generally, speed shouldn't be too fast (>3) or
        ↪ too slow (<0.3)."
    sight_range: "Sight range of the actor, in range 1-10."
    UpdateHealth: |
      (delta: number) -> void
      Update the health by `delta`.
    TimedUpdateAttackPower: |
      (delta: number, duration: number) -> void
      Update the attack power by `delta` (raw number) for `duration`
          ↪ turns. After `duration` turns, -`delta` is applied to
          ↪ restore the original value.
      Hence the net change in attack power is 0 after `duration` turns.
      This is the same for all other TimedUpdate* functions.
    TimedUpdateAccuracy: |
      (delta: number, duration: number) -> void
      Update the accuracy by `delta` (raw number) for `duration` turns.
    TimedUpdateDefense: |
      (delta: number, duration: number) -> void
      Update the defense by `delta` (raw number) for `duration` turns.
    TimedUpdateSpeed: |
      (delta: number, duration: number) -> void
      Update the speed by `delta` (raw number) for `duration` turns.
    TimedUpdateSightRange: |
      (delta: number, duration: number) -> void
      Update the sight range by `delta` (raw number) for `duration`
          ↪ turns.
    AddWaitTurns: |
      (turns: number) -> void
      Add `turns` to the wait counter of the actor, so that the actor
          ↪ will wait for `turns` turns before taking action.
```

```
          `turns` must be positive.
          Use this function very sparingly!
        RemoveBeneficialEffects: |
          (num: number) -> number
          Remove `num` random beneficial effect(s) from the actor. Return
              ↪ the number of effects removed.
        RemoveDetrimentalEffects: |
          (num: number) -> number
          Remove `num` random detrimental effect(s) from the actor. Return
              ↪ the number of effects removed.
        WithPassableCoordSelected: |
          (radius: number, fn: (coord: Coord) -> void) -> boolean
          Select an empty coordinate within `radius` distance from the
              ↪ actor's coord and apply `fn` to it.
          Return true if a coordinate is selected and false otherwise.
        WithEnemySelected: |
          (radius: number, fn: (actor: Actor) -> void) -> boolean
          Select a single enemy within `radius` distance from the actor's
              ↪ coord and apply `fn` to it.
          Return true if an enemy is selected, and false otherwise.
        GetTalentLevel: |
          () -> number
          Get the talent level of the actor for the current talent being
              ↪ defined.
          This function can only be called within the talent definition.

EffectDef:
  doc: |
      An effect definition used in NewEffect function call.
      As mentioned in the documentation for GlobalEffectTable, each Param
          ↪ type will be instantiated with a new table type.
  fields:
      name: "Name of the effect. This should roughly be a English phrase
          ↪ similar to effect_id."
      GetDescription: |
        (param: Param) -> string
        Get the description of the effect.
      OnDamageTaken: |
        (param: ParamBase, source: Actor, damage: number) -> number
        This is triggered when `param.owner` takes `damage` amount of
            ↪ damage from `source` actor.
        The return value is the modified damage value.
      OnDamageDealt: |
        (param: ParamBase, target: Actor, damage: number) -> number
        This is triggered when `param.owner` deals `damage` amount of
            ↪ damage to `target` actor.
        The return value is the modified damage value.
      OnMerge: |
        (param: ParamBase, new_param: ParamBase) -> void
        This is triggered when a new instance of the same effect is
            ↪ applied to the same actor.
        Directly modify `param` based on `new_param`.
      OnTurnEnd: |
        (param: ParamBase) -> void
        This is triggered at the end of each turn.
      OnActivate: |
        (param: ParamBase) -> void
        This is triggered when the effect is activated.
      OnDeactivate: |
        (param: ParamBase) -> void
        This is triggered when the effect is deactivated.

TalentDef:
  doc: "A talent definition used in NewTalent function call."
  fields:
```

```
name: "Name of the talent. This should roughly be a English phrase
    ↪ similar to talent_id."
GetRange: |
  (user: Actor) -> number
  Range in distance.
GetCooldown: |
  (user: Actor) -> number
  Cooldown in turns.
Do: |
  (user: Actor) -> boolean
  This is where the talent logic is implemented. `user` is the actor
      ↪ using the talent.
  Return true if the talent is successfully used, and false
      ↪ otherwise.
GetDescription: |
  (user: Actor) -> string
  Get the description of the talent.
```

We have the following global variables in the environment:

- `g_effects` of type `GlobalEffectTable`: a table containing registered effects
- `g_game` of type `GlobalGame`: the global game state object
- `g_math` of type `GlobalMath`: contains basic math functions
- `g_str` of type `GlobalString`: contains basic string functions

## C.4 Parsers for Talents and Effects

We use the following CFG to parse an effect script.

```
effect_def: data_fields_def effect_def_block
data_fields_def: "interface ParamData" "{" (key_colon_value (","
    ↪ key_colon_value)* ","?)? "}" ";"
key_colon_value: FIELD_NAME ":" "<TYPE_SPEC>"
effect_def_block: "do" "<BLOCK>" new_effect "end"
new_effect: "NewEffect" "(" "<TABLE_EXP>" ")" ";"
```

This ensures that the generated script will follow the effect template (e.g. Example B.1), where it starts by defining the parameter data type (via a special syntax `interface` ↪ `ParamData {...}`), define the effect in a do block that ends in a call to the `NewEffect` function which registers the effect in the game. After finishing parsing the effect, we register the effect in the environment in the global effect table `g_effects`, define the related methods (e.g. `g_effects.<id>.Apply`), so that later scripts can reference this new effect. The <TABLE_EXP> placeholder refers to Appendix B.2.13 and we force its type to be `EffectDef` in the API specification. All hook functions (e.g., `OnActivate`, `OnTurnEnd`) are optional fields for this table.

Next, we use the following CFG to parse a talent script which is simpler.

```
talent_def: "do" "<BLOCK>" new_talent "end"
new_talent : "NewTalent" "(" "<TABLE_EXP>" ")" ";"
```

As with effects, here we force the type of <TABLE_EXP> to be `TalentDef` in the API specification. A special API function for talents is called `GetTalentLevel`. A talent can have a number-valued level, and the inclusion of `GetTalentLevel` allows generated scripts to implement proper scaling based on the talent level, similar to in Casalini (2012).

## C.5 Provably Runtime-Error-Free Design

We detail the design choices that result in error-free execution of the generated scripts in Theorem 6.1, which is proved at the end of the section.

**Safe callback pattern to eliminate null pointers** One of the most common sources of runtime errors is dereferencing a null pointer (`nil` in Lua). We eliminate the need for null pointers by using a safe callback pattern that ensures the callback is always called with a non-nil value.

**Example C.1** (Safe callback pattern). *Instead of*

```
local actor = level.GetActorAt(coord);
if actor then
    actor.UpdateHealth(-5);
end
```

*in which the generated code might forget to check if* `actor` *is* `nil`*, we use*

```
level.WithActorAt(coord, function(actor)
    actor.UpdateHealth(-5);
end);
```

*Under the hood,* `WithActorAt` *will check if the actor exists at* `coord` *before calling the callback.*

**Limited loop iterations and lexical scoping** When translating sLua into Lua, we cap the number of iterations (to 100) in a while loop to prevent infinite loops. For loops in Lua will always terminate since the loop counters are calculated before the loop starts.

In sLua, due to the lexical scoping, the code can only reference functions that are defined lexically before the reference. In particular, we forbid recursive function calls, and local functions in the script cannot take functions as arguments due to restrictions on the type specification (Appendix B.2.6). Hence, the call graph of functions is a directed acyclic graph (DAG), and there are no infinite recursions (more formally proved at the end of this section).

**No dynamic data structures** We do not allow dynamic data structures such as arrays or tables in sLua to avoid out-of-bound access. Instead, the API functions take a callback to iterate over a range of elements, ensuring that the iteration is safe.

**Example C.2** (Range iteration with callbacks). *Imagine implementing a fireball spell that deals damage to all enemies in a ball of a radius. Instead of using an array of enemies, we use the following code:*

```
level.ProjectBall(target_coord, radius, function(actor)
    if actor.faction ~= user.faction then
        actor.TakeDamage(5);
    end
end);
```

**Cap recursion depth of effect hooks** Aside from the `Do` function in talents, the other call site to generated scripts is from the game engine calling the hook functions in effects. Since a hook function can indirectly call another hook function (for instance, a "reflective shield" effect can trigger another "reflective shield" effect on a different actor), we stop triggering new hooks if the callstack has 3 hooks already to prevent infinite recursion.

*Proof of Theorem 6.1.* First of all, because we disallow null pointers and dynamic data structures, and because our constrained decoding algorithm (Algorithm 1) ensures that only variables in scope and only predefined fields in a table will be accessed, all memory access is valid in the generated code. Furthermore, the API in Appendix C.3 is designed so that for any input arguments, as long as they type check (as guaranteed again by Algorithm 1), the API calls will not raise any runtime error.

Therefore, it remains to check generated programs will terminate in the `Do` function for each talent, and in each event hook trigger site (e.g. `OnTurnEnd`). For both cases, it boils down to checking that any generated block (Appendix B.2.1) will terminate.

We argue by induction on the program length and assume that all smaller blocks will terminate given any environment. Note that any expression should terminate because any top-level function call inside can be a single smaller statement as a prefix expression that by induction should terminate. Now consider a generated block. By looking at the CFG template for the block parser (Appendix B.2.1):

- If this is a do block, then by induction the inner block of the do block should terminate.

- If this is an if block, then the inner blocks of the if/elseif/else branches should terminate.

- If this is a for block or a while block, the inner block is guaranteed to terminate by induction. Since we limit the number of iterations in a while loop to a finite number, the overall loop should also terminate.

- If this is an assignment, local variable definition, or return statement, since expressions will terminate, these two types of statement will also terminate.

- If this is a prefix expression, by induction we may assume that it is a single function call.

  - If this function is a local variable defined in the generated code, then due to lexical scoping and the fact that the local function variable cannot have function arguments (Appendix B.2.6), the function definition block will only be able to refer to scopes before the function definition statement. Hence, by induction, function calls are guaranteed to terminate.
  - If this function is an API call, it will terminate as long as its functional arguments terminate, which is guaranteed by induction.

Hence, for all cases, generated blocks will terminate, and we are done. □

### C.6 Generating a Talent Category Using an LLM Agent

To generate all scripts in a DCI talent category from a given prompt, we use a simple LLM agent to orchestrate the generation. For each talent, we first generate a JSON (using the constrained decoding of outlines) that provides a textual description of the talent, the id of the talent, and an optional effect that should be generated first before the talent is generated. If an effect should be generated first, then we start an effect generation. Then, we do the talent generation. Each generation will be fed all the scripts generated so far so that the generated scripts will have the context to refer to the earlier scripts.

### C.7 Generating an Entire Adventure in DCI

To generate an entire adventure in DCI, we use another LLM agent to orchestrate the step-by-step generation.

- First, given an initial prompt from the player describing what kind of adventure they want to experience, we generate a JSON that gives an expanded description, a specification for the player, and the specification of all enemies. The player specification includes their name, background, and, most importantly, a description of their starting talent category. The enemy specification includes a list of five differently ranked enemies, each with their names and descriptions.

- Next, the player's starting talent category will be generated using Appendix C.6.

- Then, a talent category for all enemies will be generated, also using Appendix C.6.

- Next, we use JSON generation to assign talents to enemies. Higher-ranked enemies will have stronger talents assigned (more advanced talents or less advanced talents but at a higher talent level). We also generate different stats for enemies based on their rank.

After generating the information above using an LLM agent, we then use Wave Function Collapse[6] to generate 3-4 level maps for the adventure, and populate the enemies into various levels. The final boss will always spawn at the last level. The player is then placed randomly on the first level, and the game starts.

## C.8 Prompting for Talent Category Generation

We use the following 8 prompts to generate 8 categories of talents and effects in the experimental setup.

```
categories:
  - name: elemental_mastery
    description: Talents that channel raw elemental forces. Masters can
        ↪ unleash devastating area attacks, apply elemental effects to
        ↪ enemies, and manipulate the battlefield with fire, ice, and
        ↪ lightning.

  - name: shadow_arts
    description: Talents focused on stealth, evasion, and deception.
        ↪ Shadow artists excel at avoiding damage, repositioning in
        ↪ combat, and striking from unexpected angles.

  - name: battle_tactics
    description: Talents that emphasize weapon mastery and strategic
        ↪ combat. Tacticians can exploit enemy weaknesses, coordinate
        ↪ attacks, and control the flow of battle through positioning
        ↪ and timing.

  - name: arcane_studies
    description: Talents that manipulate magical energy and time.
        ↪ Arcanists can enhance their own abilities, create protective
        ↪ barriers, and bend reality to gain tactical advantages.

  - name: wild_survival
    description: Talents that draw power from primal instincts and
        ↪ natural forces. Survivalists excel at self-preservation,
        ↪ regeneration, and adapting to changing combat situations.

  - name: celestial_blessings
    description: Talents that channel divine energy for protection and
        ↪ healing. The blessed can shield allies from harm, restore
        ↪ health, and purge negative effects.

  - name: forbidden_knowledge
    description: Talents that sacrifice health or safety for power.
        ↪ Practitioners can drain life from enemies, enhance their own
        ↪ abilities at a cost, and inflict debilitating conditions on
        ↪ foes.

  - name: combat_engineering
    description: Talents that utilize tactical devices and battlefield
        ↪ control. Engineers can create temporary advantages through
        ↪ deployable mechanisms, control enemy movement, and set up
        ↪ devastating combo attack.
```

## C.9 Distortion of Distribution in Constrained Decoding

Distortion of distribution in constrained decoding refers to the phenomenon that the output of the LM, while satisfying the constraints, has a probability distribution different from

---

[6]https://github.com/mxgmn/WaveFunctionCollapse

the one obtained by rejection sampling (i.e., first generate an unconstrained output and reject the output until it satisfies the constraints).

As shown in the analysis by Park et al. (2024), in order to obtain the correct distribution while still performing constrained decoding, we will need to bias each token with *expected future grammaticality* (EFG), the probability that a continuation of the prefix appended with the new token results in a valid full program. EFG is intractable to compute exactly as we need to marginalize over all future continuations. Their strategy is to use a bank of past samples to compute an approximation of EFG, which relies on the bank of samples being comprehensive enough for all valid programs under constraints. While this is viable for simple grammars, in our case, the space of all semantically correct programs is immense, rendering their approximation technique unfeasible.

We comment that the infinite repetition failure cases of our method (examples given in Example C.15) are present for general constrained decoding (e.g. conforming to a JSON schema), more often with smaller language models. For example, JSON generation can output infinite whitespaces in outlines[7]. As a result, they chose the whitespace regex pattern to be a single space[8]. However, this does not preclude other repetition patterns like this one[9]. Similar issues have been reported in ollama[10] and OpenAI's JSON mode[11]. One ad-hoc solution is to penalize repetition in the last few tokens, but it does not address the root problem of distortion of distribution.

As pointed out in Section 7, our context-sensitive parser provides additional context information (e.g., the exact error type, or the regex on what needs to follow next) that could potentially be combined with post-training techniques to address this issue.

### C.10 Failure Cases for All Methods

We collect in this section representative examples for the failure examples for all methods in the talent category generation experiment from Fig. 3. In case of parsing error, we will use red to indicate the part that the sLua context-sensitive parser failed to parse, and provide the regex returned from the next_regex function that the next part needs to match against.

#### C.10.1 Unconstrained (Claude-3.5-Sonnet) with or without Reflection

**Example C.3** (Incorrect expression type). *In the example below,* power_decrease *has type* number *but was initialized to a function.*

```
local power_decrease: number = function(user)
    return 4 + 2 * user.GetTalentLevel();
end
```

*The red part failed to match regex:*

```
\s*(((((g_game|g_math)\s*\.|range\s*(;|\*|\+|\-|\/))|\(|\+|\-|\d+(\.\d+)?\
↪ s*(/|;|\*|\+|\-)))
```

**Example C.4** (Reference undefined effect id). *In the example below,* shield_wall *is an effect id that was not defined in the environment so far. The set of defined effect ids are shown in the regex below.*

```
g_effects.shield_wall.Apply(user, {...
```

*The red part failed to match regex:*

---

[7] https://github.com/dottxt-ai/outlines/issues/691

[8] https://dottxt-ai.github.io/outlines/latest/reference/generation/json/

[9] https://github.com/dottxt-ai/outlines/issues/1131

[10] https://github.com/ollama/ollama/issues/2577

[11] https://community.openai.com/t/streaming-generation-stops-and-prints-many-whitespaces/876038/12

```
\s*((blinded|combo_points|confusion|damage_shield|enraged|poisoned|
    ↪ stunned|vulnerable|wounded)\s*\.)
```

**Example C.5** (Hallucination of API of hooks). *In the example below, the LLM thought* Actor *class has* OnDamageTaken, *which does not. This is a hook function that cannot be accessed by the scripts by design to prevent faulty logic like the one here that attempts to replace the hook function with something else. Because there is no function of type* (Actor, number)-> number *in scope, in the regex, the only acceptable string is the* function *keyword which will lead to a function definition.*

```
local old_damage_taken: (Actor, number) -> number = user.OnDamageTaken;
user.OnDamageTaken = function(source, amount)
    if source.faction ~= user.faction then
        g_effects.heat_resonance.Apply(source, {
            damage = damage
        }, resonance_duration);
    end
    return old_damage_taken(source, amount);
end;
```

*The red part failed to match regex:*

```
\s*(function)
```

*In the example below, the LLM thought* Level *class has* SetImpassableToEnemy *function which is not defined in the API. All available functions of* Level *can be seen in the regex.*

```
g_game.level.SetImpassableToEnemy(param.data.coord, true);
```

*The red part failed to match regex:*

```
\s*((GetPushedCoord|IsPassable|MoveActor|ProjectActorsShuffled|
    ↪ ProjectBall|ProjectLine|WithActorAt|WithRandomEmptyCoordInRadius)\
    ↪ s*\()
```

**Example C.7** (Hallucination of API in effect parameter data). *In the example below, the LLM thought the parameter instance* param *has* first_trigger *field defined from earlier. However, the correct way to access* first_trigger *is via* param.data.first_trigger.

```
interface ParamData {
    first_trigger: boolean,
};
do
    NewEffect({
        name = "Untargetable",
        beneficial = true,
        detrimental = false,
        OnDamageDealt = function(param, target, damage)
            if damage > 0 and param.first_trigger then
```

*The red part failed to match regex:*

```
\s*(((data|owner)\s{0,50}(==|\.|\s+and|\s+or|\s+then|\~=)|duration\s
    ↪ *(<=|<\ |==|>=|>\ |\*|\+|\-|\/|\~=)))
```

**Example C.8** (Wrong syntax to define a function). *In the example below, it uses a wrong syntax to define a local function. The sLua specification requires the format* local GetHealAmount: ↪ (Actor)-> number *instead. The generated code tried to mimic the syntax of luau (Roblox, 2019) but we do not allow such syntax.*

```
local function GetHealAmount(user: Actor): number
    return g_math.Floor(GetHealingFactor(user) * user.attack_power / 10);
end;
```

*The red part failed to match regex:*

```
\s*([a-zA-Z_]\w{0,49}\s*:)
```

### C.10.2    Unconstrained (Qwen2.5-32B-Coder) with or without Reflection

Compared to Claude-3.5-Sonnet, the open-source Qwen2.5-32B-Coder makes significantly more syntax or obvious errors. Errors similar to the ones in Appendix C.10.1 are still present and will not be repeated here.

**Example C.9** (No semicolon after parameter data definition). *Our effect parser requires having a semicolon after* ParamData *interface definition but this example failed to do so.*

```
interface ParamData { }

do
```

*The red part failed to match regex:*

```
\s*(;)
```

**Example C.10** (Not following the effect template). *For effect parser to parse successfully, the effect definition must be wrapped inside a* do *block. The following code failed to do so.*

```
interface ParamData {
    accuracy_increase: number,
};

NewEffect({
```

*The red part failed to match regex:*

```
\s*(do)
```

**Example C.11** (No type specification for variables). *In local variable definitions, we require a* : *after the variable name followed by a type specification. However, in this example, the generated code failed to add a type specification.*

```
local combo_point_effectiveness = user.GetTalentLevel() * 1.5;
```

*The red part failed to match regex:*

```
\s*([a-zA-Z_]\w{0,49}\s*:)
```

**Example C.12** (Add type specification when it is not needed). *For an inline function expression, since we already know the type of the function, we disallow type specification. Yet in the example below, the generated code added the extra type specification.*

```
local CheckBehindEnemy: (Actor, Coord) -> boolean = function(user,
    ↪ target_coord)
    local user_coord: Coord = user.coord;
    local direction_x: number = user_coord.x - target_coord.x;
    local direction_y: number = user_coord.y - target_coord.y;
    local behind_coord: Coord = { x = user_coord.x + direction_x, y =
        ↪ user_coord.y + direction_y };

    return g_game.level.IsPassable(behind_coord) and
        ↪ g_game.level.WithActorAt(behind_coord, function(actor: Actor)
        return actor.faction ~= user.faction;
    end);
end;
```

*The red part failed to match regex:*

```
\s*([a-zA-Z_]\w{0,49}\s*\))
```

**Example C.13** (Add comments which are not allowed). *We explicitly mentioned in the sLua language specification that comments (e.g. lines starting with --) are not allowed. Yet the LLM can still output comments.*

```
do
    NewEffect({
        name = "Earth Spike Land",
        beneficial = false,
        detrimental = true,
        OnActivate = function(param)
            -- Implement logic to make the tile impassable for enemies
        end,
```

*The red part failed to match regex:*

```
\s*(((g_effects|g_game|g_math|g_str|param)\s*\.|do\s+|end|for\s+|if\s+|
    ↪ local\s+|param\s*(=|\.)|return(;|\s+)|while\s+))
```

**Example C.14** (Reference undefined local variable). *In the example below, the LLM wrongly assumed* user *is in scope because many previous lines all refer to* user. *However, it is not defined.*

```
do
    local GetDamagePercent: (Actor) -> number = function(user)
        return 1.0 + 0.1 * user.GetTalentLevel();
    end;

    local GetStunDuration: (Actor) -> number = function(user)
        return 2;
    end;

    local GetShockDuration: (Actor) -> number = function(user)
        return 3;
    end;

    local range: number = 6 + user.GetTalentLevel();
```

*The red part failed to match regex:*

```
\s*((((GetDamagePercent|GetShockDuration|GetStunDuration)\s*\((|(g_game|
    ↪ g_math)\s*\.)|\((|\+|\-|\d+(\.\d+)?\s*(/|;|\*|\+|\-)))
```

### C.10.3 Ours (Qwen2.5-32B-Coder) with or without Reflection

All failure cases using our method (Algorithm 1) are due to nontermination of the generation. We show a few failure cases here.

**Example C.15** (Intend for comments but parsed as operators). *As we have seen in Example C.13, despite prompt instruction, Qwen2.5-32B-Coder can still output comments sometimes. While constrained decoding prevents comments from being generated most of the time, the first character of a usual comment starting symbol can be interpreted as a numerical operator and be accepted, cornering the generation to generate infinite repetitions of junk text as the following example shows:*

```
do
    local GetMovementBoost: (Actor) -> number = function(user)
        return 0.5 + 0.1 * user.GetTalentLevel()
    /

    - 0.5 + 0.1 * user.GetTalentLevel() - 1.0
    - 1.0 + 0.1 * user.GetTalentLevel() - 1.5
```

```
- 1.0 + 0.1 * user.GetTalentLevel() - 1.5 + 0.5 *
    ↪ user.GetTalentLevel() - 1.5
- 1.0 + 0.6 * user.GetTalentLevel() - 1.5
- 0.6 * user.GetTalentLevel() - 0.5

- 1.0 + 0.1 * user.GetTalentLevel() * 2
- 1.0 + 0.2 * user.GetTalentLevel()

- 0.8 - 0.1 * user.GetTalentLevel()

- 0.5 * user.GetTalentLevel()

- 1.0 + 0.1 * user.GetTalentLevel()

- 1.0 + 0.1 * user.GetTalentLevel()
- 2 * user.GetTalentLevel() / 10 + 1
... (same pattern repeats infinitely)
```

*In this case, since we allow expressions to cross lines (we treat all whitespace characters in the same way), because the first line did not end up in* ; *, the parser considers the* / *character a division operator, whereas the LLM intended to generate* // *which marks the beginning of a line comment for C-like languages. In future lines, the LLM intends to go back to Lua-style comments with* -- *, but the second* - *was rejected by the parser (recall we disallow consecutive* -- *), so it spirals into weird generation that repeats infinitely. This failure case can be treated if we only allow spaces inside an expression, so a newline character will not be accepted until a* ; *is parsed.*

**Example C.16** (Poor quality generated code spirals indefinite future generation). *In the rather lengthy example below, we see that when the earlier generated code quality is poor, as shown here with lots of local functions that didn't do anything interesting, the later generation starts to copy this pattern and keeps going on.*

```
do
    local GetHealAmount: (Actor) -> number = function(user)
        return 2 + user.GetTalentLevel() * 1;
    end;

    local GetDuration: (Actor) -> number = function(user)
        return 5 + user.GetTalentLevel() * 2;
    end;

    local GetAccuracyBoost: (Actor) -> number = function(user)
        return 1 + user.GetTalentLevel() * 0.5;
    end;

    local ApplyNatureEmbrace: (Actor, number) -> boolean =
        ↪ function(user, duration)
        local param_data: NaturesEmbraceParam = {
            duration = duration,
            owner = user,
            data = { accuracy_boost = GetAccuracyBoost(user) },
        };
        g_effects.natures_embrace.Apply(user, param_data.data,
            ↪ param_data.duration);
        return true;
    end;

    do
        local talent_def: (Actor) -> boolean = function(user)
            local heal_amount: number = GetHealAmount(user);
            local duration: number = GetDuration(user);
            user.UpdateHealth(heal_amount);
            ApplyNatureEmbrace(user, duration);
            return true;
        end;
```

```
local talent_desc: (Actor) -> string = function(user)
    local heal_amount: number = GetHealAmount(user);
    local duration: number = GetDuration(user);
    local accuracy_boost: number = GetAccuracyBoost(user);
    return "Heals the user for " .. g_str.from_num(heal_amount)
        ↪ .. " health every turn for " ..
        ↪ g_str.from_num(duration) .. " turns. Grants Nature's
        ↪ Embrace, increasing accuracy by " ..
        ↪ g_str.from_num(accuracy_boost) .. " for the
        ↪ duration.";
end;

local talent_def_table: (Actor) -> boolean = function(Actor)
    return talent_def(Actor);
end;

local talent_desc_table: (Actor) -> string = function(Actor)
    return talent_desc(Actor);
end;

local primal_healing_talent: (Actor) -> boolean = function(user)
    return talent_def(user);
end;

local talent_range: (Actor) -> number = function(user)
    return 0;
end;

local talent_cooldown: (Actor) -> number = function(user)
    return 20 - user.GetTalentLevel() * 3;
end;

local talent_name: string = "Primal Healing";

local primal_healing_talent_def: (Actor) -> boolean =
    ↪ function(user)
    return talent_def(user);
end;

local primal_healing_talent_desc: (Actor) -> string =
    ↪ function(user)
    return talent_desc(user);
end;

local primal_healing_talent_range: (Actor) -> number =
    ↪ function(user)
    return 0;
end;

local primal_healing_talent_cooldown: (Actor) -> number =
    ↪ function(user)
    return 20 - user.GetTalentLevel() * 3;
end;

local talent_def_final: (Actor) -> boolean = function(user)
    return talent_def(user);
end;

local talent_desc_final: (Actor) -> string = function(user)
    return talent_desc(user);
end;

local talent_range_final: (Actor) -> number = function(user)
    return 0;
```

```
        end;

        local talent_cooldown_final: (Actor) -> number = function(user)
            return 20 - user.GetTalentLevel() * 3;
        end;
        ... (same pattern repeats infinitely)
```

**Example C.17** (High-probability code rejected midway causing constrained decoding to self-imitate). *In the code below, in the first return statement, after the* `return` *keyword, initially the LLM wants to generate something else that was rejected by the parser, and then it chose to continue with* `return true and`, *which does not do anything meaningful. Then it learns its own pattern and gets stuck in an infinite self-imitation process.*

```
(omit the beginning)
if g_math.Random() < 0.5 then
    g_game.level.WithRandomEmptyCoordInRadius(target.coord,
        ↪ GetSpectralHandTeleportRange, function(coord)
        g_game.level.MoveActor(target, coord);
        g_game.level.ProjectActorsShuffled(coord, GetRange(user),
            ↪ function(new_target)
        if new_target.faction ~= user.faction then
            return true and
                g_game.level.WithRandomEmptyCoordInRadius(new_target.coord,
                    ↪ GetRange(user), function(final_coord)
                return g_game.level.WithActorAt(final_coord,
                    ↪ function(final_target)
                if final_target.faction ~= user.faction then
                    return true and
                        attacks_remaining > 0 and
                        turns_remaining > 0 and
                        g_game.level.WithRandomEmptyCoordInRadius(final_coord,
                            ↪ GetRange(user),
                            ↪ function(last_coord)
                        return
                            ↪ g_game.level.WithActorAt(last_coord,
                            ↪ function(last_target)
                        if last_target.faction ~=
                            ↪ user.faction then
                        return true and
                        attacks_remaining > 0 and
                        turns_remaining > 0 and
                        g_game.level.WithRandomEmptyCoordInRadius(last_coord,
                            ↪ GetRange(user),
                            ↪ function(final_coord_2)
                        return
                            ↪ g_game.level.WithActorAt(final_coord_2,
                            ↪ function(final_target_2)
                        if final_target_2.faction ~=
                            ↪ user.faction then
                        return true and
                        attacks_remaining > 0 and
                        turns_remaining > 0 and
                        g_game.level.WithRandomEmptyCoordInRadius(final_coord_2,
                            ↪ GetRange(user),
                            ↪ function(final_coord_3)
                        return
                            ↪ g_game.level.WithActorAt(final_coord_3,
                            ↪ function(final_target_3)
                        if final_target_3.faction ~=
                            ↪ user.faction then
                        return true and
                        attacks_remaining > 0 and
                        turns_remaining > 0 and
                        ... (same pattern repeats infinitely)
```

