# OpenReview forum: "Correctness-Guaranteed Code Generation via Constrained Decoding"
_colmweb.org/COLM/2025/Conference — COLM 2025_

### Official Review · Reviewer_b3mK · 2025-05-12

**Rating:** 6
**Confidence:** 3
**Ethics Flag:** 1

**Summary:**

The paper tackles the problem of generating one-shot-correct code with large language models (LMs). It introduces context-sensitive constrained decoding driven by a dynamic Tree-of-Parsers (ToP). Each node in the tree is an interactive parser for a small, modular CFG that is instantiated with live scope and type information, so the partial program is semantically valid after every step. To guide the LM, the root node emits a regular expression that satisfies a non-extensible-match property, guaranteeing the LM can decide exactly when to stop its next token burst. A “token-healing” patch handles sub-word tokens that straddle regex boundaries, while a hybrid strategy interleaves unconstrained generation to amortise DFA-compilation cost. Algorithm 1 orchestrates these pieces into an incremental decoding loop . The framework is demonstrated on sLua, a strongly-typed Lua dialect. With off-the-shelf Qwen 2.5-32B-Coder, the method reliably produces game-mechanic scripts. Overall, the work shows that by combining modular context-sensitive parsing with constrained decoding, LMs can be pushed beyond syntax to deliver provably correct code suitable for embedded production settings.

**Questions To Authors:**

1. Would it be possible to add results on a well-known public code-generation benchmark to demonstrate how the method performs in a setting that readers can readily compare against?
2. Could you include a brief ablation showing how much each of your three key contributions; non-extensible regex, token-healing, and hybrid fallback; contributes to success rate and latency?
3. Roughly how many lines or hours did it take to build the modular CFGs, environment objects, and placeholder logic for sLua? What would it take to replicate this for other languages and what are the requirements?

**Reasons To Accept:**

1. **Interesting Problem:** The paper is one of the first to show how to get an LM to emit code that is semantically and even runtime correct on the first try, turning LMs from copilots into safe autonomous coders.
2. **Technical Novelty:** Proposes a new approach combining Tree-of-Parsers (ToP), Non-extensible-match oracle + token healing, and Hybrid decoding.
3. **Guarantees:** By combining ToP with a carefully restricted, strongly-typed Lua dialect, the authors prove termination, absence of nil-dereferences and out-of-bounds errors for every generated script.

**Reasons To Reject:**

1. **Extremely narrow language scope:** All results are for sLua, a highly restricted subset of Lua. Because the safety proof and linear-time guarantee rely on those restrictions, the method is untested on mainstream languages with richer features.
2. **Heavy manual grammar engineering:** There is a huge upfront cost involved of engineering the grammar (refactoring the entire language grammar into dozens of modular CFG templates, etc.). This upfront cost undermines practicality; MGD [A] and similar monitor approaches need only a language server which are ubiquitous.
3. **Performance Bottleneck:** Compiling the token-accepting DFA with the Willard & Louf indexer is “about 9x slower than the LM calls” in total wall-time . The paper suggests using a faster (but closed-source) transducer or falling back to unconstrained generation, which weakens guarantees. A 9x slowdown is hard to justify for interactive IDE settings or real-time game scripting.
4. **Guarantees depends on API design:** Proof of runtime safety depends on a hand-crafted game API that is total, side-effect-free and uses “safe callback” wrappers . Outside such sandboxed APIs the theorem does not hold. Limits the relevance of the results to real-world code that calls arbitrary libraries.
5. **Limited and Unconventional Evaluation:** Only 8 talent-category prompts and 10 runs each are scored, and quality is judged by another LLM (Claude 3.5) rather than human experts or functional tests. No comparison to compilation-rate or pass@k on public benchmarks. The evidence may be too anecdotal and raises questions on general applicability.
6. **Interesting but Weaker Advancement over Prior Art:** Compared to Monitor-Guided Decoding [A], ToP achieves stronger guarantees but only by redesigning the language and API; it is unclear that the core ideas cannot be merged into an existing monitor architecture with less effort. It may be argued that the contribution-to-engineering-effort ratio is not compelling.

# References
* [A] Monitor-Guided Decoding of Code LMs with Static Analysis of Repository Context, NeurIPS 2023.

---

> ### Author Response · Authors · 2025-05-29
>
> Dear reviewer,
>
> Thank you for your insightful and critical comments. For concerns regarding the narrow language scope and the manual engineering effort, please refer to the **Restriction to sLua and its engineering effort** section and the **Adaptation to other languages** section in the common response. For concerns regarding the performance bottleneck, we have improved our regex compilation method with a >3x speedup using a new lazy indexing algorithm, which makes it no longer a bottleneck. Please refer to the  **New speedup on the regex compilation step** section for details.
>
>
> *Comparison with [Agrawal et al.], Monitor-Guided Decoding (MGD)*
>
> Thank you for pointing out this reference which we will add in our revision in addition to a proper discussion in the revised paper.
>
> Compared to our approach, [Agrawal et al.] considers a narrower setting of semantic correctness: MGD is only concerned with the correctness on a small subset of language features, instead of on the entire program as in our paper or in [Poesia et al]. Their concrete method only handles derefencing an object and making sure correct identifiers would follow. This is far from making sure the entire program is semantically correct (as evidenced in the sub-80% compilation rate in their reported results). As argued in the common response, achieving 100% semantic correctness requires a revamp of the language design with a tailored parser. Hence, we strongly disagree with the reviewer’s assessment that our “advancement is weak over prior art.”
>
> Moreover, MGD does not guarantee correctness even on dereferencing an object. Here’s a counterexample to their strategy proposed at the end of Section 2: If the LLM has a very long token that starts with a valid identifier and matches $w \cdot E \cdot \Sigma^*$ but ends in an invalid suffix, their algorithm can still accept this token, resulting in an incorrect program. In comparison, our token-healing algorithm is designed for handling such a case, where this last token will be rolled back before starting with the next regex.
>
>
> *Would it be possible to add results on a well-known public code-generation benchmark to demonstrate how the method performs in a setting that readers can readily compare against?*
>
> Since we propose a new problem (runtime-correctness guaranteed code generation in a constrained environment), we are not aware of any public benchmark focusing on this aspect, as most are targeted at generic problem solving. Our work introduces a novel problem: runtime-correctness guaranteed code generation within a constrained environment. To our knowledge, existing public benchmarks primarily focus on general problem-solving and do not specifically address this aspect. Therefore, we are unaware of any current benchmarks directly comparable to our proposed problem.
>
>
> *Could you include a brief ablation showing how much each of your three key contributions; non-extensible regex, token-healing, and hybrid fallback; contributes to success rate and latency?*
>
> The non-extensible regex and token-healing are essential parts of our algorithm - they cannot be swapped out. Without non-extensible regexes, we do not have a way to make constrained decoding terminate for regexes with wildcards (Ex. 3.1). Without token-healing, tokens cannot cross regex boundaries.
>
> We provided an ablation study on the regex compilation step. We improved our previous hybrid approach (based on [Willard & Louf]) to using a new lazy indexing algorithm, getting a 3.8x speedup. This is also much faster than [Koo et al.] which we previously mentioned might have an improvement over [Willard & Louf] but our experiment shows otherwise.

---

> > ### Author Response · Authors · 2025-05-29
> >
> > *Roughly how many lines or hours did it take to build the modular CFGs, environment objects, and placeholder logic for sLua? What would it take to replicate this for other languages and what are the requirements?*
> >
> > It takes a considerable amount of effort to design the modular CFGs, environment objects, and placeholder logic for sLua. The total number of lines of parser code is not excessive—only 2k lines of Python code. Thanks to our modular design, we offload most of the parsing logic to the CFG parser generator (Lark in this case), and the primary remaining work is on injecting context and applying the look-ahead strategy (described at the end of Section 4) to ensure non-extensibility. Once the sLua ToP is implemented, we can extend it to generate code following arbitrary API functions and custom types with just a few lines of code.
> >
> > As mentioned in the **Restriction to sLua and its engineering effort** section of the common response, we emphasize that for the runtime-critical applications that we considered where no programmer is in the loop, the actual choice of language is not that important as long as it has a sufficiently rich feature set and that we have a way to execute the generated programs. In addition, we can translate sLua to any modern language by translating the resulting sLua AST, if we want to use a different interpreter/compiler. A future research question is then how to further expand the feature set of sLua while maintaining the existing guarantees on linear-time parsing speed and the runtime-correctness guarantee.

---

> > > ### Comment · Reviewer_b3mK · 2025-06-05
> > > **Response**
> > >
> > > ## Narrow language scope (sLua only)
> > >
> > > Response points to a "Restriction to sLua" section and asserts that (i) sLua can be translated to mainstream languages and (ii) the real-time settings they care about don’t need richer features.
> > >
> > > **Assessment:** No new experimental evidence or prototype for a second language is provided, so the validity of the approach beyond sLua remains speculative. Concern remains largely open.
> > >
> > > ## Heavy manual grammar engineering
> > > Response says the ToP implementation is ~ 2k LoC, most logic lives in Lark CFGs, and adding new APIs is "a few lines".
> > >
> > > **Assessment:** The 2k LoC number is useful, but effort is still non-trivial; Partially addressed.
> > >
> > > ## Performance bottleneck (9x DFA cost)
> > >
> > > Response introduces a lazy-indexing compilation giving a 3.8 × speed-up, claiming the step is "no longer a bottleneck".
> > >
> > > **Assessment:** Addressed.
> > >
> > > ## Limited / unconventional evaluation
> > >
> > > Response argues that no public benchmark targets “runtime-correctness-guaranteed” generation, so they cannot compare.
> > >
> > > **Assessment:** The request here was to use any widely used benchmark (e.g. HumanEval compilation rate) to gauge generality of supported sLua feature set. This request is declined, so the weakness stands.
> > >
> > > ## Ablation of key contributions
> > >
> > > Response claims non-extensible regex and token-healing are indispensable so cannot be ablated; only shows speed ablation for regex compilation.
> > >
> > > **Assessment:** Addressed. Albeit, it is quite unclear why these aspects are indispensible. For example, what if there was no token healing, would every program simply fail?
> > >
> > > The rebuttal adequately covers the MGD comparison and provides quantitative detail on grammar size and a partial speed fix. It does not convincingly resolve the broader generality, evaluation, and ablation concerns, and the guarantees still hinge on a highly curated language + API bundle. However, in light of the additional analysis, I will bump my score slightly.

---

### Official Review · Reviewer_GqNm · 2025-05-12

**Rating:** 8
**Confidence:** 2
**Ethics Flag:** 1

**Summary:**

The authors present an interesing work about LLM-based semantically correct code generation. They present a novel algorith to generate correctness-guaranteed programs via constrained decoding. To do that, the authors use a context-sensitive parser built as a tree of parsers.

The authors make a demonstration of their system using an ad-hoc strongly-typed version of Lua (named sLua) and a roguelike game as a demonstrator. New game mechanics are added on-the-fly by generating scripts with new effects and talents.

**Questions To Authors:**

* References: in general, some of them contain the usual lowercase errors.
* In (Hopcroft, 2001), complete the name of the author.

**Reasons To Accept:**

* A really hot topic.
* Well written, well organized, "easy to read" (taking into account that is a complex reading).
* Novelty.

**Reasons To Reject:**

I see no reasons, but I'm not sure if a so-long appendix is acceptable.

---

> ### Author Response · Authors · 2025-05-29
>
> Dear reviewer,
>
> Thank you for your positive assessment. We will fix the reference problems that you pointed out in the revised version. Please refer to the common response that addresses other reviews’ concerns, and a new table showing a speedup of our new lazy indexing algorithm.

---

> > ### Comment · Reviewer_GqNm · 2025-06-05
> > **No more comments, I'm done.**
> >
> > It's ok. I have no more comments nor questions.

---

### Official Review · Reviewer_YK73 · 2025-05-18

**Rating:** 8
**Confidence:** 4
**Ethics Flag:** 1

**Summary:**

- Introduces a system that uses a Tree of Parsers (ToP) to guide LLMs in generating correct code.
- Designs sLua, a safe, strongly typed subset of Lua tailored for runtime correctness.
- Uses modular CFGs and interactive parsers to incrementally parse and validate code.
- Compiles regexes into DFAs to filter out invalid tokens before the LLM samples.
- Ensures semantic correctness by tracking variable types and scopes in a dynamic environment.
- Demonstrates the system in a roguelike game, generating talents and effects on the fly.

**Questions To Authors:**

- How scalable will the approach be to languages that will not satisfy the assumptions made in the paper?
- Is the idea that we will create a specific language version/parser to leverage this for new languages?
- Can this also help high resource languages like Java?

**Reasons To Accept:**

- Ensures both syntactic and semantic correctness during code generation.
- Guarantees runtime safety through language and API design.
- Modular and extensible parsing framework (Tree of Parsers).
- Works with off-the-shelf LLMs without needing model retraining.
- Demonstrated in a real-world application (game scripting) with strong results.

**Reasons To Reject:**

- Relies on a custom, restricted language (sLua), limiting generalizability.
- Requires manual effort to define and maintain modular CFGs.
- Constrained decoding can still fail due to non-termination or repetition.
- Complexity of the system may hinder adoption or replication.
- Limited evaluation outside the game scripting domain.

---

> ### Author Response · Authors · 2025-05-29
>
> Dear reviewer,
>
> Thank you for your positive review. For concerns about sLua and its engineering effort, please refer to the **Restriction to sLua and its engineering effort** section and the **Adaptation to other languages** section in the common response. We provide additional answers to the specific questions you raised below.
>
> *How scalable will the approach be to languages that will not satisfy the assumptions made in the paper? Is the idea that we will create a specific language version/parser to leverage this for new languages? Can this also help high resource languages like Java?*
>
> As discussed in the common response, sLua is carefully designed to achieve the linear-time parsing guarantee that we have. If we are okay with slower parsing speed, we could include more features into the language design. The framework of ToP—using modular CFGs to represent the language— is still applicable, except the size of the tree can be much bigger.
>
> On the other hand, our target application is for runtime critical scenarios where there is no programmer involved. As such, the exact choice of the programming language is less relevant, as long as we have a compiler/interpreter to execute the program. We think sLua already provides a rich set of features (e.g. conformation to any prescribed API with any user-defined fixed-size types) and is sufficient for a lot of tasks such as in generative game scripting. On the other hand, as discussed in the common response, since the ToP gives us the AST of the program, we can also translate the resulting sLua to another host language like Java/C++. Such translation requires a bit of work compared to translating to Lua (which is what we did in our paper).

---

> > ### Comment · Reviewer_YK73 · 2025-06-02
> > **Thanks for the response**
> >
> > Thank you for the response! This clarifies my questions.

---

### Official Review · Reviewer_kWD4 · 2025-05-20

**Rating:** 7
**Confidence:** 4
**Ethics Flag:** 1

**Summary:**

This paper proposes Tree-of-Parsers, a constrained decoding algorithm that guarantees the semantic correctness of generated programs. ToPs is built to maintain the non-extensibility assumption and extends CFGs from guaranteeing syntax to also guaranteeing semantic correctness. For languages that follow certain assumptions, ToPs can parse them incrementally with linear time. Under some stricter assumptions, ToPs can guarantee there is no runtime error in its generated code.

**Questions To Authors:**

1. How easy is it to integrate this into existing efficient constrained decoding libraries such as XGrammar (https://github.com/Dan-wanna-M/formatron) or formatron (https://github.com/Dan-wanna-M/formatron)?

2. Is this the best we can do with constrained decoding or are there more properties that we can guarantee with it?

3. How different is your algorithm from Mündler et al?

Reference

Mündler, Niels, et al. "Type-Constrained Code Generation with Language Models." arXiv preprint arXiv:2504.09246 (2025). https://arxiv.org/abs/2504.09246

**Reasons To Accept:**

1. Well written paper that clearly describes the algorithm and how it improves constrained decoding, with demonstrative examples and good explanation on the motivation of every part. I'm convinced that ToPs guarantees semantic-correctness more efficiently than existing algorithms such as Poesia et al (2021).

2. Carefully designe experiments on sLua that demonstrate the necessity of ToPs for customized APIs and its runtime-error-free property under certain assumptions.

**Reasons To Reject:**

1. Experiments seem a bit too concentrated on sLua. Could you provide some justification on how it works on other languages and in other domains?

2. Lack of comparison with either existing constrained decoding algorithms or ablative baselines. What if some essential parts were removed from ToPs or some less-effective constrained decoding algorithm were used? For example, one advantage the authors claims is linear complexity incremental parsing instead of whole program verification at each token. How much of an improvement in efficiency does that give us?

---

> ### Author Response · Authors · 2025-05-29
>
> Dear reviewer,
>
> Thank you for your positive comments. For a discussion of why we restrict sLua and not consider other languages, please refer to the **Restriction to sLua and its engineering effort** section and the **Adaptation to other languages** section in the common response. For ablative results on different strategies of regex compilation step, please refer to the **New speedup on the regex compilation step** section in the common response. We comment that other components of our algorithm are specially designed (in particular, ToP) and no alternative exists. Moreover, it is not possible to swap out parts of ToP because it essentially defines the sLua language.
>
>
> *Integration to XGrammar or Formatron?*
>
> Both XGrammar and Formatron focus on context-free grammars, in particular on JSON outputs. For instance, XGrammar requires a pushdown automaton formulation that is not applicable to the context-sensitive setting that we study. We would like to emphasize that our setting is sufficiently different from these works and requires a careful design of the grammar and a curated selection of language features from ground up.
>
>
> *Is this the best we can do with constrained decoding or are there more properties that we can guarantee with it?*
>
> Providing runtime correctness guarantees using constrained decoding is in a sense the ultimate goal from a compiler/static analysis point of view. However, as mentioned in our limitation section, the program being correct does not mean the program is necessarily high quality, as constrained decoding leads to distortion of distribution (App. C.9). It is an interesting future direction to investigate how other techniques, such as post-training, can improve the quality of the generated program in addition to the correctness.
>
>
> *How different is your algorithm from Mündler et al?*
>
> Thank you for pointing out this highly relevant concurrent work. We detail the differences below and we will revise our related works section to include this work.
>
> - First, we focus on generating runtime-correct sLua code, whereas their goal is to reduce compilation error for TypeScript without guarantees to compile.
> - Their prefix automaton is similar in spirit to our ToP, where every reachable state can lead to an accepting state. Both their prefix automaton and our ToP are built recursively based on the target language’s CFG. Both employ a type reachability algorithm (their Figure 3 vs our Section B.2.7). However, theirs is at a low-level, where the state transitions are on a character level. We operate on a terminal level of context-free grammar. Because of their lower-level design, their method likely requires a significantly higher amount of engineering (they reported >10k lines of Python code compared to our 2k lines of Python code for parsing sLua). Because of this, we expect our framework to be more extendable to other languages.
> - We support type declaration in the generated code (e.g. generated effect types can later be used). This is not supported by Mündler. For a predefined type (e.g. `Actor`), it seems like Mündler’s approach additionally requires manual effort to build a prefix automaton for each of these types, whereas our approach is automatic. While they do support more features such as forOf loops, dynamic data structures (arrays, sets, and maps), these features can also be incorporated into sLua if we are only concerned with semantic correctness. Since our goal is on runtime correctness, we forbid dynamic data structures while providing workarounds such as safe callbacks.

---

> > ### Comment · Reviewer_kWD4 · 2025-06-05
> >
> > Thank you for the response. I've read it and decided to keep my score.

---

### Author Response · Authors · 2025-05-29
**Common response to reviewers**

Dear reviewers,

Thank you for your valuable feedback on our work. We respond here to the common questions shared by multiple reviewers. We also highlight a new speedup on the regex compilation step thanks to a new lazy indexing algorithm that we added to the pipeline since the submission.

**Clarification on the scope of the problem at hand**

- We are considering 1-shot correctness of generation for runtime critical scenarios like embodied agents or generative game mechanics. This is a drastically different problem from developing a more reliable coding copilot that helps programmers write code. Our application requires generated code that is semantically and syntactically correct from the outset, without human intervention.
- The difference between being correct 80% of the time vs. 100% of the time is enormous. The former only requires handling specific cases (type checking, or dereferencing an object), while the latter requires a fundamental rethinking of language design.
- Since this is a new problem that has not been studied properly before, no existing benchmarks or methods exist.


**Restriction to sLua and its engineering effort**

- Because of the extra challenging requirement of the problem at hand, we designed a new language with restricted features to enable us to satisfy the goal. In runtime critical scenarios, since no programmer is in the loop, the exact choice of the language matters less as long as it can be executed.
- We carefully engineered the ToP for sLua with a linear-time parsing speed guarantee, believing that it serves as a prime example of how to design a programming language amenable to LM generation with semantic/runtime guarantees.
- Thanks to our modular design and the offloading of parsing challenges to a CFG parser generator, the engineering effort on sLua is significant but manageable (2k lines of Python code). We plan to open source sLua and its ToP. The same design principles can be applied to designing other languages (which can be a subset of features from a well-known language like Java).

**Adaptation to other languages**

- An sLua program can be readily translated to a mainstream host language by transforming the resulting sLua AST to the AST of the host language. Or one can write their own sLua interpreter in any host language (or using a Lua interpreter in the host language given our provided sLua -> Lua translation).
- On the other hand, a ToP can be built for another language that e.g. resembles Java or C++ following a similar recipe, but the language features will need to be restricted to attain correctness guarantees. We are not sure whether this will unlock better parsing guarantees or generation quality compared to sLua, and we think this is a good future direction. We chose Lua as the reference language because of its simplicity and the application domain (games) we considered.

**New speedup on the regex compilation step**

We implemented a new lazy indexing algorithm with adaptive rejection sampling that **achieves a 3.8x total speedup** over [Willard & Louf] that we used at the time of submission. This new algorithm extends the constrained/unconstrained hybrid approach and lazily builds a token-accepting DFA with rejection sampling. We have also ablated the different compilation strategies, including [Willard & Louf] and our own implementation of [Koo et al.] that did not provide their implementation.

| | Unconstrained | Willard & Louf | Koo et al. | Ours (lazy indexing) |
|---|---:|---:|---:|---:|
| **tokens/sec** | 18.89 | 2.04 | 1.47 | 7.64 |
| **compilation/total** | 0% | 73.35% | 79.41% | 4.98% |

We see that our new lazy indexing algorithm significantly reduces the compilation time and improves the overall tokens/sec. We have revised our paper to reflect this change.

---

### Decision · Program_Chairs · 2025-07-08

**Decision:**

Accept

**Comment:**

The reviewers praised the paper for its clear algorithm description, well-designed experiments, and strong results. The proposed Tree of Parsers (ToPs) framework is modular and extensible, ensuring both syntactic and semantic correctness during code generation. The authors have also thoroughly addressed reviewers' concerns and questions, providing comprehensive and thoughtful discussions with additional information.

Although the approach appears to be limited to the new sLua language designed by the authors, certain components and ideas can be applied to future extensions. As a result, the paper still provides a meaningful contribution, as the reviewers unanimously agree that this paper should be accepted.

In preparation for the next version, the authors are highly encouraged to incorporate their responses to reviewers' concerns and questions, which will undoubtedly further improve the paper's quality.

[Automatically added comment] At least one review was discounted during the decision process due to quality]